# Integrating nutrition and physical activity promotion: A scoping review

**Laura Casu** [1]☺*, **Stuart Gillespie**[2]☺, **Nicholas Nisbett**[1]☺

**1** Institute of Development Studies (IDS), University of Sussex, Brighton, United Kingdom, **2** International Food Policy Research Institute (IFPRI), Washington, DC, United States of America

☺ These authors contributed equally to this work.
* L.Casu@ids.ac.uk

## Abstract

### Background

This paper investigates actions that combine nutrition and physical activity which hold potential for 'double duty action' to tackle multiple forms of malnutrition simultaneously. Expanding on previous research on single component actions, we assessed the state of the literature to map integrated interventions across the life course to analyse potential double duty effects, and identify knowledge gaps and needs for future design, implementation, evaluation and research for effective double duty action.

### Methods

A scoping review of peer-reviewed and grey literature was conducted to explore the pathways that extend from combined physical activity and nutrition promotion interventions, with potential synergistic effects on outcomes other than obesity. Electronic databases were searched for studies published between 1 January 2010 and 31 January 2020. Out of 359 articles retrieved, 31 peer-reviewed and 5 grey literature sources met inclusion criteria. Findings from 36 papers reporting on 34 interventions/initiatives were organised into 6 categories, based on implementation across multiple stages of the life course. Double-duty potential was assessed through a further stage of analysis.

### Findings

This review has identified actions that hold potential for tackling not only obesity, but healthy diets, sedentary behaviour and quality of life more generally, as well as actions that explicitly tackle multiple forms of malnutrition. Importantly, it has identified crucial gaps in current methods and praxis that call for further practice-oriented research, in order to better understand and exploit the synergistic effects of integrated interventions on outcomes other than obesity.

### Conclusions

Findings from across implementation settings suggest that even in situations where interventions are aimed at, or framed in terms of, obesity prevention and control, there are

**Data Availability Statement:** All relevant data are within the manuscript and its Supporting Information files.

**Funding:** Financial support for this study was provided by the CGIAR Fund Donors through the

CGIAR Research Program on Agriculture for Nutrition and Health (A4NH), led by IFPRI. We would like to thank all funders who supported this research through their contributions to the CGIAR Trust Fund: https://www.cgiar.org/funders/. CGIAR Fund Donors had no role in study design, data collection and analysis, decision to publish, or preparation of the manuscript.

**Competing interests:** The authors have declared that no competing interests exist.

unexploited pathways for broader outcomes of relevance to nutrition and health and wellbeing more generally. Future design and evaluation of multisectoral approaches will benefit from an explicit framing of interventions as double-duty oriented.

## 1. Introduction

Malnutrition, in all its forms, is the largest single risk factor for the global burden of disease [1]. Every country is affected, and many countries are dealing with a "double burden" characterized by the coexistence of undernutrition with overweight, obesity or diet-related non-communicable diseases (NCDs) [2]. Alongside pervasive problems of undernutrition, the prevalence of overweight/obesity is rising dramatically, with 39% adults overweight or obese in 2016 [3]. Obesity is increasing in most countries, in both urban and rural settings, and across socio-economic levels—raising the risk of NCDs such as type 2 diabetes, hypertension, dyslipidemia, and various cancers [3]. 41 million of the world's 57 million deaths (71%) in 2016 were due to NCDs, with the highest burden falling on low and middle-income countries [4, 5]. No country has yet succeeded in reversing this trend.

This paper responds to recent calls to review efforts to create enabling environments for 'double-duty actions' (DDAs) [6]. In their potential for preventing or reducing several forms of malnutrition simultaneously by targeting common or shared drivers of both undernutrition and overweight [6], double-duty actions hold great promise, especially in populations where multiple forms co-exist. The acknowledgement of involuntary exposure to underlying and structural determinants informs the wide range of strategies to promote wellbeing in multiple settings and across the lifespan. As a necessary step in developing an integrated agenda for addressing the root causes of malnutrition at all stages of the life course, this review focuses on integrated interventions/initiatives that combine both nutrition and physical activity, which have yet to be reviewed through a double-duty lens. This will expand on previous research on single component actions, with the anticipation that such integrated initiatives are likely to be framed mainly as obesity prevention and control programmes. Yet, being implemented primarily in countries with a high dual-burden or in specific settings (such as locations with poor socio-economic indicators and vulnerabilities) with higher risks of multiple forms of malnutrition and food insecurity, they have high potential for addressing all forms of malnutrition and related comorbidities.

Undernutrition and overweight/obesity are systemically connected to broader socio-political determinants of disease [7, 8] and act synergistically (or 'syndemically') [9] where they cluster in poorer and more marginalised populations. Such synergistic effects are important. The UN's 2018 State of Food Security and Nutrition in the World report [10] employed a conceptual framework (page 30) that highlighted different pathways from inadequate food access to multiple forms of malnutrition. In addition to food and nutrition insecurity, the key non-nutritional pathway to overweight and obesity is mediated by poor mental health. This is based on growing evidence of associations between food insecurity and anxiety, stress and depression that are independent of other indicators of low socio-economic status in both resource-rich and resource-poor settings [11–13]. Stress brought on by food insecurity may cause non-homeostatic eating and may lead to the selection of 'comfort' foods, or highly palatable foods that are rich in fat, sugar, and sodium [14, 15]. Finally, physical inactivity, in and of itself, is a major health issue [16–22]. Insufficient physical activity is one of the leading risk

factors for death worldwide [23]. Globally, physical inactivity is estimated to account for between 6–10% of ischaemic heart disease, stroke, diabetes, and breast and colon cancer [24].

In this context, a key question is—are there benefits to integrating approaches to address both poor diet and physical inactivity simultaneously? Are there potential synergies to be exploited? Does *integration* of physical activity and nutrition promotion hold potential for generating effects on malnutrition that are more than simply additive, particularly in the context of these shared/syndemic broader determinants and contexts?

Fig 1 is a modified form of the framework used in the State of Food Security and Nutrition in the World 2018 Report (page 30) [10] that highlights the pathways of interest that extend from combined physical activity and nutrition promotion interventions. Because many physical activity and nutrition promotion programmes include an element of school feeding (e.g. breakfast clubs), we also add this as an additional/optional intervention component. The diagram illustrates the pathways to two primary sets of outcomes (reduced obesity/overweight and increased dietary diversity/reduced micronutrient deficiencies), as well as one related set of intermediate outcomes (physical and mental wellbeing). While we do not hypothesise a link between physical activity and dietary diversity, the point of the diagram is to show that even in situations where interventions are aimed at, or framed in terms of, obesity prevention and control, there are unexploited pathways for broader outcomes of relevance to nutrition and health and wellbeing more generally. It is these 'additionalities', and their potential relevance for double duty action, that are the focus of this review.

In pursuit of these aims, we conducted a scoping review to a) map interventions across the life course to identify actions that combine nutrition and physical activity with potential double duty effects; b) identify whether outcomes of relevance are being appropriately measured; c) assess whether and to what extent these outcome measures can be used to inform DDA analysis; and d) identify knowledge gaps and needs for future design, implementation, evaluation and research for effective double duty action.

## 2. Methods

### 2.1. Literature search strategy

The search strategy was designed to capture both peer-reviewed and grey literature—the latter deemed essential, given that much knowledge and evidence on integrated nutrition and physical activity interventions derives from innovation in practice.

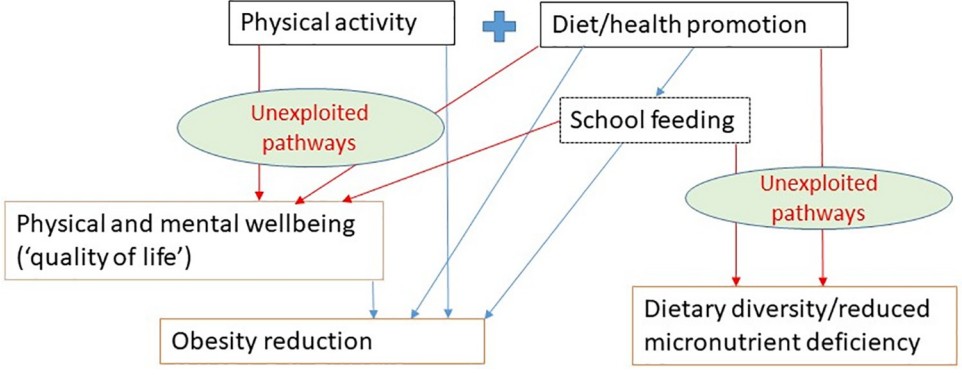

**Fig 1. Conceptual framework showing unexploited pathways.**

For peer-reviewed literature, the databases PubMed and Web of Science, and studies from the Public Library of Science (PLoS) and *The Lancet*, were searched for relevant intervention studies. A targeted search based on search terms 'physical activity' AND 'nutrition' AND 'intervention' OR 'initiative' AND 'double burden' OR 'double duty' was carried out, with adapted variants for each database where appropriate. The syntax was built on Pubmed ["physical activity" OR exercise [MeSH] AND nutrition OR Diet, Food, and Nutrition [MeSH] OR dietary AND intervention OR initiative AND "double burden" OR "double duty"]. Syntax variations included search terms 'physical activity' (to include relevant Exercise [MeSH] terms for Pubmed) AND 'nutrition' (to include Diet, Food and Nutrition [MeSH] OR dietary OR nutrition (free text word) for Pubmed and TS = nutrition OR diet* for Web of Science) AND intervention* OR initiative* (wildcards) with applied filters for Timespan = 2010–2020 and Language (English). To be included in the review, studies must have a) addressed interventions or initiatives to promote physical activity *and* nutrition; b) been published from 1 January 2010 to 31 January 2020; c) provided a description of the intervention; and d) reported awareness, knowledge, behavioural, physiological, environmental or organisational change outcomes.

The following were excluded: a) studies that focused on synergistic elements without being integrated in interventions with a nutrition and physical activity component (e.g. single policy, taxation, changes to the built environment only); b) studies on actions that did not *combine* nutrition and physical activity but focused on either one of these elements individually; c) articles not associated with an intervention or community-wide initiative, such as studies on prevalence, correlates, or determinants, and d) articles in languages other than English.

For grey literature, conventional review methods for searching, appraising, managing, and synthesising the evidence base can be adapted [25–28]. A grey literature search plan was thus developed to incorporate five different searching strategies: a) grey literature databases, b) snowballing from available grey literature, c) customised Google search engines, d) targeted websites, and e) consultation with relevant experts. These complementary strategies were used to minimise the risk of omitting relevant sources. Since abstracts are often unavailable in grey literature documents, full-text screening was employed in the initial search and text was then coded into categories for future retrieval. Targeted searches were conducted to identify actions combining nutrition and physical activity. We included interventions and initiatives for which some degree of monitoring and evaluation was available. Structured searches of relevant websites and search engines were complemented by citation tracking and consultation with relevant experts, which helped to follow leads to additional relevant material.

## 2.2. Extraction and analysis

We used a standard extraction template to extract information on study and intervention characteristics. Data were collected on author(s), year of publication, study setting, study design and sampling, and purpose of study. Intervention-specific data were extracted on implementing organisation, type and description of intervention, outcomes, key findings and constraints to effectiveness (see Table 1). Appraisal of the quality of included studies, conducted separately following a standard assessment [29], additionally involved examination of the extent of participation, empowerment, multisectorality and sustainability of programmes. A narrative synthesis was undertaken in which insights were mapped with regard to delivery platforms that applied to different stages of the life cycle, in consideration of the recognised need to act beyond the first 1,000 days and across all life-stages to address root causes of malnutrition and prevent co-morbidities from escalating later in life [30, 31].

**Table 1. Summary of integrated interventions.**

| First author/year | Platform/Programme | Country/Study design | Sampling | Intervention description | Outcomes |
|---|---|---|---|---|---|
| Bock (2014) | Health service<br><br>Family focus<br><br>The Health Initiative Program for Kids (HIP Kids) | Canada<br><br>1—year longitudinal pilot intervention study | Children and adolescents with obesity referred by family physicians /paediatricians in London, Ontario and surrounding communities. Eligible participants (aged 8–17 with primary obesity & at least 1 caregiver willing to participate) prospectively enrolled in order of referral. | Bi-weekly education sessions for three months and monthly education sessions for nine months (with dietician, social worker, fitness specialist). | Improvements in the participants' BMI z-scores and QoL following a 1-year intervention, while other adiposity-related measures of body composition remained unchanged. |
| Bolton (2017) | Community<br><br>The Health-Promoting Communities: Being Active Eating Well initiative | Australia<br><br>A mixed method and multilevel quasi-experimental evaluation | Children and adolescents (cross-sectional) and adults (longitudinal) with data collected pre- and post-intervention. Intervention (n = 2408 children and adolescents from 18 schools, n = 501 adults from 22 workplaces) and comparison groups (n = 3163 children and adolescents from 33 schools, n = 318 adults from seven workplaces). | Promotion of healthy eating, physical activity and stronger, healthier communities. | Gains in community capacity, but few impacts on environments, policy or individual knowledge, skills, beliefs and perceptions. Relative to comparison group, one community achieved lower prevalence of overweight/obesity, lower weight, waist circumference and BMI ($P<0.005$); one community achieved higher level of healthy eating policy implementation in schools; two communities achieved improved healthy eating-related behaviours ($P<0.03$); one community achieved lower sedentary behaviours; and one community achieved higher levels of physical activity in schools ($P<0.05$). All effect sizes were in the small-to-moderate range. |
| Cheadle (2018) | Community<br><br>Kaiser Permanente's Healthy Eating Active Living Zones Initiative | USA<br><br>Population-dose approach and pre/post surveys to assess impact of policy, program, and environmental change strategies. | 12 low- income communities in Kaiser Permanente's Northern and Southern California Regions. | Each HEAL Zone is a small, low-income community of 10,000 to 20,000 residents with high obesity rates and other health disparities. Community coalitions planned and implemented multidimensional strategies in each community. Over 230 community change strategies implemented over 3 years. | Positive population-level results were observed for higher- dose strategies, particularly those targeting youth physical activity. Higher-dose strategies were more likely to be found in communities with the longest duration of investment. Policy, environmental, and programmatic changes as well as efforts to build community capacity to sustain strategies and make changes in the future. |
| Colchamiro (2010) | Early care<br><br>Massachusetts Women, Infants, and Children (WIC) Program | USA<br><br>Two qualitative research methods, laddering and projective techniques, used in focus group activities to uncover emotional pulse points common to WIC mothers. | 32 WIC mothers from 6 participating agencies took part in focus groups to determine the acceptance of emotion-based nutrition services. 16 in-depth interviews conducted with WIC participants to determine specific opinions regarding 30 emotion-based nutrition education materials and facilitated group discussions. | Behaviour change through the Touching Hearts, Touching Minds initiative. Nutrition education materials and facilitated group counselling techniques developed through the project use the power of parent-identified emotional "pulse points" to guide WIC participants to practice positive eating and physical activity behaviours. | Increased opportunities to transform the nutrition counselling relationship between WIC families and WIC staff through provision of materials and strategies achieved through state-wide implementation. Changes in nutrition education in Massachusetts duplicated across the country, and beyond national boundaries in numerous venues with finalised Touching Hearts, Touching Minds materials translated into Spanish, Portuguese, French, Russian, Chinese, and Vietnamese. |

*(Continued)*

**Table 1.** (Continued)

| First author/year | Platform/Programme | Country/Study design | Sampling | Intervention description | Outcomes |
|---|---|---|---|---|---|
| Correa (2010) | Community<br><br>CAN DO Houston (Children and Neighbors Defeat Obesity; la Comunidad Ayudando a los Niños a Derrotar la Obesidad) | USA<br><br>Process evaluation | Houston Sunnyside and Magnolia Park neighbourhoods identified through previous data on health and wellness from stratified cluster sampling. | A community-based, childhood obesity prevention initiative in 2 low-income neighbourhoods in Houston. Coordination of various activities to promote healthy living, including after-school programs, grocery store tours, wellness seminars, cooking classes, and staff wellness clubs, on the basis of input from community members. | Residents' uptake of additional opportunities to participate in physical activity and nutrition education in their communities. Outputs based on 5 evidence-based target behaviours and key messages that served as focal point of initiatives: eat 9 fruits and vegetables daily, eat breakfast daily, limit screen time to no more than 2 hours daily, engage in moderate to vigorous physical activity for 60 minutes daily, and spend 60 minutes of uninterrupted family time together daily. Participants identified lack of physical activity as the primary barrier. |
| Cradock (2016) | OST (out-of-school time)<br><br>The Out of School Nutrition and Physical Activity (OSNAP) initiative | USA<br><br>Cluster-randomised controlled trial with matched program pairs. | 20 after-school programs in Boston, Massachusetts. All children 5 to 12 years old in participating programs were eligible for study inclusion. | The OSNAP initiative works with after-school programmes to improve nutrition and physical activity–related practices, environments and policies, using a socioecological model and a community-based participatory research approach. Nutrition-related activities and added physical activity component. The 2 physical activity goals were (1) to include 30 minutes of moderate, fun physical activity for every child every day (including outdoor activity, if possible) and (2) to offer 20 minutes of vigorous physical activity (VPA) 3 times per week. | Improved nutrition practices, environments, and policies. Participating programs did not allot significantly more time for physical activity. Existing program time made more vigorously active for children, but no effect on moderate to vigorous physical activity. Change in duration of physical activity opportunities during program time did not differ between conditions. Change in moderate to vigorous physical activity minutes accumulated by children during program time did not differ significantly by intervention status. Total minutes per day of vigorous physical activity, vigorous physical activity minutes in bouts, and total accelerometer counts per day increased significantly during program time among intervention participants compared with control participants. |
| Davis (2013) | Preschool<br><br>CHILE: Child Health Initiative for Lifelong Eating and Exercise; preschool intervention for obesity prevention in Head Start | USA<br><br>5-year efficacy trial | All children enrolled in Child Health Initiative for Lifelong Eating and Exercise (CHILE), 16 Head Start centres. | Evidence-based intervention to prevent obesity in children enrolled in 16 Head Start (HS) Centres in rural communities. HS is a comprehensive program that provides educational, social, health, nutritional, and other services to children from low-income households. CHILE uses a socioecological approach to improve dietary intake and increase physical activity. The intervention includes: a classroom curriculum; teacher and food service training; family engagement; grocery store participation; and health care provider support. Five levels of influence: individual or intrapersonal, interpersonal, organisational, community, and public policy. | Increased opportunities to taste new fruits and vegetables and sustain healthy habits, dietary behaviour change achieved through nutrition curriculum and foodservice component. At least 60 minutes of unstructured (free-play) and at least 60 minutes of structured (adult-led) physical activity each day. Increased opportunities for HS teachers to sustain changes through provision of the tools, equipment and lessons needed to incorporate an additional 30 minutes of physical activity per day into classroom time. |

(*Continued*)

**Table 1.** (Continued)

| First author/year | Platform/Programme | Country/Study design | Sampling | Intervention description | Outcomes |
|---|---|---|---|---|---|
| Folta (2015) | OST<br><br>Healthy Kids Out of School (HKOS) initiative, volunteer-Led Out-of-School Time Programs | USA<br><br>Case study | 6 focus groups with program leaders from Boy Scouts, YMCA, 4-H, and Pop Warner Football and Cheer. Half-day roundtable discussions with program administrators working at the state or regional level. Additional outreach through key informant interviews (by telephone) with program leaders and administrators. | The Healthy Kids Out of School (HKOS) initiative developed evidence-based, practical guiding principles for healthy snacks, beverages, and physical activity. | Dissemination of the HKOS guiding principles in large and complex OST (out-of-school time) organisations was best accomplished by using implementation strategies that were customised, integrated, and aligned with goals and usual practices. Dissemination strategies that put the burden on individual leaders to champion adoption of guiding principles in the organisation was found to be problematic, especially in programs that depend heavily on unpaid volunteers for administration and operations (e.g. 4-H and Pop Warner Football and Cheer). |
| Griffin (2018) | Higher risk group<br><br>My Quest, using text messaging to improve behaviours among low-income women | USA<br><br>1-group, pre- to post-test study design | 55 Alabama counties (84% rural) with high rates of poverty, overweight/obesity, and chronic diseases. | Short texts (n = 2-3/d) provided health tips, reminders, and goal-setting prompts. Weekly electronic newsletters provided tips and recipes. Participant self-monitored body weight weekly. | Improved dietary and physical activity behaviours and food environment. Increased dietary and physical activity goal setting. Reduced body weight. |
| Hinkle (2018) | Community<br><br>Food & Fitness (F&F) Community Partnerships | USA<br><br>Data from interviews, conducted over the phone, which included semi-structured, open-ended questions. Review of data from annual project reports, meeting minutes, and partnership notes summarizing youth activities and involvement. | 73 youth interviewed between 2008 and 2016 by evaluators from University of Michigan. Youth were selected to be interviewed by the project director of each partnership. | Food & Fitness (F&F) initiative, a 9-year community-based intervention, to ensure that all children have equitable access to healthy food and built environments that promote safe physical activity. The youth engagement component focused on strategies and structures that would support a model framework for youth involved in F&F community partnerships, incl. skill -building, technical assistance and leadership development beyond adult community members to include youth. | Over the course of 9 years, provision of training to thousands of young people, leadership and technical skills and uptake of collaborative efforts to sustain change in communities with inequities which result in poor health outcomes for children. Behaviour changes depending on level of engagement of youths participating in activities, increased advocacy to ensure that everyone in the community had access to affordable, healthy, locally grown food and safe places to be physically active. Increased advocacy within homes was additional effect of behaviour change communication training aimed at wider community. |
| Irwin (2012) | Community- School-Home<br><br>Get Fit with the Grizzlies initiative | USA<br><br>Survey research over first 4 years. Matched pre/post test design. McNemar's test for significance (<05) was applied to measure correct answers pre and post. | Matched pre/post test design (n = 2210) in randomly chosen schools (n = 18) from all elementary schools in the Memphis area. | 2006 the Memphis Grizzlies (the city's National Basketball Association (NBA) franchise), launched 'Get Fit with the Grizzlies', a 6-week, curricular addition focusing on nutrition and physical activity for 4th & 5th grades in Memphis City schools. The health-infused mini-unit was delivered by the physical education teachers during their classes. National and local sponsors with matching 'Get Fit' objectives were solicited to fund the program. In 2010–11 school year, 'Get Fit' evolved into a new program called 'Healthy Home Court' which added a breakfast component at high schools where data indicated greater need. | Overall successful multi-year initiative. Mixed results across venues. Increased opportunities for health knowledge acquisition and health behaviour change. Increased physical activity in all intervention high schools. Breakfast attendance numbers showed different results across high schools. |

*(Continued)*

**Table 1.** (Continued)

| First author/year | Platform/Programme | Country/Study design | Sampling | Intervention description | Outcomes |
|---|---|---|---|---|---|
| Jung (2018) | Community<br><br>Family focus<br><br>Healthy Together (HT) physical activity and healthy eating family program | Canada<br><br>Mixed-methods approach. Sources of data included archival records, interviews and surveys. | 10 sites implemented the 5-week program. 39 staff members and 277 program participants (126 caregivers [Mage = 35.6] and 151 children [Mage = 13]). | Healthy Together (HT) is a family centred education program developed by The Bridge Youth and Family Services to promote healthy weights in children from vulnerable populations (i.e., rural, remote, northern, Aboriginal and multicultural communities across Canada). | Changes in caregivers' confidence that they could engage in healthy eating practices. Changed children's knowledge of desired min. of physical activity. Overall Healthy Together represents a feasible community-based intervention but had limited effectiveness. No changes in shopping practices. No changes in knowledge of physical activity requirements for adults or children, fruit and vegetable consumption requirements for adult and children, or knowledge of screen time limits from pre to post-program, or from pre to 6-month follow-up. No differences in knowledge were found within individual sites. No changes in parental social support provided for physical activity or healthy eating. No changes in children's knowledge of the daily fruit and vegetable requirements or sugary drinks. No changed knowledge of screen time recommendations. |
| Lang (2017) | Workplace<br><br>The National Healthy Worksite Program | USA<br><br>2013 and 2015 survey data from the National Healthy Worksite Program, a Centres for Disease Control and Prevention (CDC)-led initiative to help workplaces implement health-promoting interventions. Multilevel regression models. | 41 employers completed the CDC Worksite Health Scorecard to document organisational changes. 825 employees provided data to evaluate changes in their health and attitudes. | In 2011, the Centers for Disease Control and Prevention (CDC) launched the national healthy worksite program (NHWP). The goal was to assist approximately 100 employers in implementing comprehensive workplace health programs with evidence-based and promising health promotion and disease prevention interventions that would improve health outcomes to reduce chronic disease. | Successfully implemented interventions to improve physical activity and nutrition. Targeting of and uptake by a large number of small employers (less than 100 employees), provision of training and tools for employers to select, tailor, and implement their own interventions from an extensive list of evidence-based interventions actualised across sites. Onsite support and training to employers during planning, implementation and evaluation of programs. Changes in awareness of employers, improved skills, new knowledge and skills applied by employers, more direct control over planning and implementation assumed by employers. Increased opportunities for employees' healthy eating and physical activity, and incentives for reducing tobacco use. Changes in awareness and behaviour of employees. |

(*Continued*)

**Table 1.** (Continued)

| First author/year | Platform/Programme | Country/Study design | Sampling | Intervention description | Outcomes |
|---|---|---|---|---|---|
| Linton (2014) | Community<br><br>The San Diego County Childhood Obesity Initiative (COI) | USA<br><br>Theory-based evaluation of advocacy, a novel approach to community youth obesity prevention. Data from adult leader surveys, decision-maker interviews, and baseline-midline-endline surveys with youth. | At various times during the 2-year evaluation period, 21 groups were formed, and all consented to participate in the evaluation. | A program of the non-profit Community Health Improvement Partners, COI is a public–private partnership for reducing and preventing childhood obesity through policy, systems, and environmental change. San Diego State University collaborated with the San Diego County Childhood Obesity Initiative to evaluate Youth Engagement and Action for Health! (YEAH!), a youth advocacy project to engage youth and adult mentors in advocating for neighbourhood improvements in physical activity and healthy eating opportunities. | 20 groups engaged in advocacy. Most (n = 13) conducted school assessments, 5 assessed parks, and 2 assessed outdoor advertising. Fast food, store, and street advertising assessments were each conducted by 1 group. On the basis of these assessments, 8 groups identified issues with playgrounds, parks, and recreation facilities, and 5 groups identified issues relating to school food. All 20 groups engaged in advocacy with decision makers; 19 groups completed in-person presentations or meetings. 11 groups reported a change implemented as a result of their advocacy, 4 groups reported changes pending, and 5 groups saw no change as a result of their efforts. Advocacy resulted in changes such as the addition of salad bar at high school, addition of exterior lighting at community centre so youth could walk to and from centre at night, addition of female-only swim time at YMCA, facilitating participation by young Muslim women. |
| Lyn (2013) | Child care centre<br><br>Wellness policy and training program in 24 child care centres in Georgia | USA<br><br>Environment and Policy Assessment and Observation instrument to identify changes to foods served, staff behaviours, and physical activity opportunities. | For-profit centres (n = 14) and non-profit centres (n = 10) were included in the program; 4 centres offered the Head Start program. The centres served a total of 2,042 children aged 2 to 5 years (range, 40–245 children per centre during the program). | The Child Nutrition and Women, Infants, and Children Reauthorization Act of 2004 included a provision requiring local education agencies participating in national nutrition programs (ie, National School Lunch Program, School Breakfast Program, and Special Milk Program) to adopt and implement local wellness policies in schools. The goal of the mandate was to promote wellness and health by focusing on nutrition education, physical activity, and other school-based activities to create healthy school environments. From February 2010 through April 2011, a program was conducted in southwest Georgia that focused on training caregivers in adoption and implementation of 6 wellness policies designed to improve nutrition and physical activity practices. | Significant improvements to total nutrition (P < .001) and physical activity scores (P < .001) were observed. Centres significantly improved the physical activity environments by enhancing active play (P = .02), the sedentary environment (P = .005), the portable environment (P = .002), staff behaviour (P = .004), and physical activity training and education (P < .001). Significant improvements were found for the nutrition environment (P < .001), and nutrition training and education (P < .001). |

(*Continued*)

**Table 1.** (Continued)

| First author/year | Platform/Programme | Country/Study design | Sampling | Intervention description | Outcomes |
|---|---|---|---|---|---|
| Madsen (2015) | School<br><br>The Healthy Schools Program (HSP) | USA<br><br>Used state-wide body mass index (BMI) data to measure changes in prevalence of overweight obesity. | Preferentially targeted school districts in northern and southern California with at least 50% of students eligible for free or reduced-price meals (FRPM), selecting districts from urban, suburban, and rural regions. Analyses were limited to the 281 public schools with BMI and demographic data available. Of the 6,350 non-HSP California schools with similarly complete data, 709 control schools were selected by using propensity-score 'nearest neighbour' matching. | The Alliance for a Healthier Generation's Healthy Schools Program (HSP) is a national evidence-based obesity-prevention initiative aimed at providing the schools in greatest need with onsite training and technical assistance (TTA) and consultation with national experts (HSP national advisors) to create sustainable healthy change in schools' nutrition and physical activity environments. | Findings are based on analysis of data collected annually from 5th-, 7th-, and 9th-grade students to determine whether enrolling in the HSP's onsite intervention reduced prevalence of overweight and obesity in intervention schools (n = 281) versus propensity-score matched control schools (n = 709), and whether increasing exposure to the program (TTA and contact with HSP national advisors) was associated with reductions in the prevalence of overweight and obesity. HSP appears to be an important means of supporting schools in reducing obesity. Although participation in HSP alone was not sufficient to improve weight status in California schools, there was a clear dose-response relationship to the program. HSP serves as an effective model for addressing childhood obesity among engaged schools. |
| MCA-Mongolia (2013) | Community<br><br>Millennium Challenge Account–Mongolia 5-year NCD prevention programmes | Mongolia<br><br>Overview of nation-wide policy implementation of activities aimed at prevention of NCDs. | Population level. Countering the default approach of the Mongolian social security system, i.e. looking after people when they are sick, rather than preventing illness and injury through healthy lifestyles, MCA embraced the health promoting workplace concept as an important point of access for tackling non-communicable diseases and injuries. | Two main focuses: 1. Behaviour change (e.g. physical exercise and healthy diet) and reduction of risk factors (e.g. alcohol and tobacco consumption), with an emphasis on enabling Mongolians to take responsibility for their own health. 2. Encourage short-term applied research on cost-effective ways to control NCDs and accidents. (e.g. stretching exercise appears on employees' computer screens at set times during the day). | Dietary behaviour change, increased incentives for practising healthy behaviours, reduction of risk factors in work environments, increased opportunities for exercise during working hours. Effective building of networks and partnerships for promoting and sustaining change. Set up the Health Promoting Workplace Network, following meetings with stakeholders incl. Ministry of Health, Mongolian Employers' Federation (MONEF), Chamber of Commerce and Trade, M. Trade Union Federation, and M. Public Health Association. Created business case for healthy workplaces by analysing temporary disability allowances paid out to employees of various agencies by both employers and the social insurance fund. Showed significant opportunity to decrease financial and service costs incurred by employers through workplace health promotion. Introduction of related changes in food industry in parallel with MCA-Mongolia's behaviour change campaigns. |

(*Continued*)

**Table 1.** (Continued)

| First author/year | Platform/Programme | Country/Study design | Sampling | Intervention description | Outcomes |
|---|---|---|---|---|---|
| McAuley (2010) | Community<br><br>The APPLE (A Pilot Program for Lifestyle and Exercise) project | New Zealand<br><br>The marginal costs of the project in 2006 prices were estimated and compared with the kilograms (kg) of weight-gain prevented for children in the intervention relative to the control arm. The children's health-related quality of life (HRQoL) was also measured using the Health Utilities Index (HUI). | All children enrolled in the seven primary schools servicing the intervention (n = 4) and control (n = 3) communities were invited to participate. Response rates at each measurement point were uniformly high, ranging from 81–89% in control schools and 85–92% in intervention schools and subjects were predominantly white (82.6%, 16.5% Maori and <1% Pacific Islander). | 2-year controlled community-based obesity prevention initiative utilising activity coordinators (ACs) in schools and nutrition promotion in New Zealand children (5–12 years). Provision of community ACs at each school, managed by a program coordinator, to encourage all children to be a little more physically active every day. Activity-based interventions included teacher-aid resource called "Snacktivity" to facilitate short bursts of activity during class time, provision of extra sports and activity equipment available at interval and lunch times. Nutrition-based interventions included supplying schools with a cooled water filter and provision of free fruit for a 6-month period. Targeted reductions in sugary drinks and increased fruit and vegetable intakes including "APPLE Bites" (community-based resource highlighting ideas, recipes, hints and tips for being more active and eating well at home), science lessons and innovative card game simulating completing a triathlon. | Total project cost was NZ $357,490, or NZ$1,281 per intervention child for 2 years (NZ $1 = US$0.67 = UK pound 0.35 = EUR euro 0.52). Increased variety and opportunities for physical activity at interval, lunchtime and after school beyond what was being provided, with emphasis on less traditional sports and more lifestyle-based activities. Increased opportunities and uptake of short bursts of activity. Increased intake of fruit and vegetables. Decreased consumptions of sugar-sweetened beverages. Weight z-score was reduced by 0.18 (0.13, 0.22) units at 2 years and 0.17 (0.11, 0.23) units at 4 years in intervention relative to control children. Mean HUI values did not differ between intervention and control participants. The reduction in weight z-score observed is equivalent to 2.0 kg of weight-gain prevented at 15 years of age. The relatively simple intervention approach employed by the APPLE project was successful in significantly reducing the rate of excessive weight gain in children, with implementation costs of NZ $664–1,708 per kg of weight-gain prevented over 4 years. |
| McDavid (2016) | Child care centre<br><br>Growing Fit: engaging early care environments in preventing obesity | USA<br><br>First report related to this project, describes the training and its dissemination between January and December 2015. | 103 early childcare educators from 39 early childcare education centres (22 individual childcare systems) from 19 counties in Georgia. 15 systems completed pre/post test assessment. | The Georgia Department of Public Health, Georgia Shape—the Governor's Initiative to prevent childhood obesity, and HealthMPowers Inc., created the Growing Fit training and toolkit to assist early childhood educators in creating policy, systems, and environmental changes that support good nutrition and physical activity. | Increased awareness and knowledge of early child educators, increased awareness and knowledge of parents, improved motivation of staff and parents, changed attitudes in relation to food, tv time and active lifestyle for kids. Increased opportunities for parents and children to spend time with the family and others in community gardens. Lessons learnt from the first year of training include the need for more robust assessment of adoption and implementation of policy, systems, and environmental changes in trained centres. |
| McIsaac (2017) | School<br><br>Health-Promoting Schools program in Nova Scotia, Canada | Canada<br><br>Cross-sectional evaluation of a natural experiment. | Population-based survey of students in grades 4–6 (about 9–12 years old) and their parents in the Tri-County Regional School Board (TCRSB) in Nova Scotia, Canada. | A Health Promoting Schools (HPS) approach aims to make schools a healthy place through a holistic approach that promotes a supportive 'school ethos' and emphasizes improvements in physical, social, and emotional well-being and educational outcomes. | Did not find significant differences between HPS and non-HPS schools. Results highlight the complexity of evaluating HPS in the real world. Differences in impacts measured. The findings suggest the potential role of a supportive school ethos for well-being in school. |

(*Continued*)

**Table 1.** (Continued)

| First author/year | Platform/Programme | Country/Study design | Sampling | Intervention description | Outcomes |
|---|---|---|---|---|---|
| Meinen (2018) | Child care centre | USA | Quantitative data collected from respondents (n = 25) involved in a 2015 survey of healthTIDE stakeholders (n = 310). Set of 5 items to rate level of agreement, based on 3-point scale, with statements about collective impact practices. Purposive sampling of small group of active leaders, representing a range of roles, affiliations, and years of involvement, and in-depth, semi-structured individual interviews (n = 7). | The Wisconsin Early Childhood Obesity Prevention Initiative, established in 2007, seeks to address and prevent obesity in the early care and education system through nutrition and physical activity environmental and policy changes. The collaborative includes professionals from 3 state of Wisconsin Departments, the University of Wisconsin-Extension, the University of Wisconsin-Madison, and public health and early care and education organisations. | Progress in establishing the conditions for collective impact. Improved capacities and improved formal and informal practices for setting of common agenda, aligning and coordinating partner activities, and promoting communication among Initiative leaders. Achieved significant policy, systems, and environment changes since inception. |
| | The state-wide Wisconsin Early Childhood Obesity Prevention Initiative | Mixed methods case study | | | |
| Miller (2018) | School | USA | All grantee schools in ChildObesity180 and FUTP60 grant programmes. | Let's Move! Active Schools (LMAS), now Active Schools, is a national initiative in the United States (US) that aims to engage schools to increase students' opportunities to be physically active, as an addition to existing nutrition components, school meals programmes and food availability in schools, within the broader framework of Fitness, Sports & Nutrition Guidelines already in place. Additional financial support for schools in the form of small grants [namely, LMAS-partner grant from ChildObesity180 or Fuel Up to Play 60 (FUTP60)], at a national scale, for schools to make changes that support PA throughout the day. Promotion of healthy lifestyles via messaging, implementing classroom PA breaks, and providing PA before and after school. | Increased opportunities for students to be physically active and increased physical activity levels. The findings show that small grants, at a national scale, for schools to make changes that support physical activity throughout the day can shift physical activity practices over the course of a school year. |
| | Let's Move! Active Schools (LMAS) | ChildObesity180 and FUTP60 grantee schools completed nine common questions, before and after receiving the grants to assess progress in implementing practices for PE and PA. D-in-D and comparison between grant programmes. | | | |
| Mukhina (2014) | School | Russia | School population of more than 13,000 children attending public schools in the Leningrad (Lomonosovskii District), Vladimir, and Novgorod regions. | The program serves more than 13,000 children attending public schools in the Leningrad (Lomonosovskii District), Vladimir, and Novgorod regions. BeHealthy provides funding for schools and comprehensive educational materials to help schoolchildren develop habits of healthy nutrition and physical activity, as well as consulting and expert support for school staff and other key stakeholders. The program brings in experts on program implementation and training for teachers. Curriculum support also includes printed and Web-based healthy lifestyle educational materials on best practices and positive experience, as well as meetings and conferences with school representatives and local authorities. | Reported inconclusive findings based on 4 interrelated activities: conducting lessons for schoolchildren on healthy nutrition, with emphasis on breakfast; healthy cooking lessons with children; cultivating nutritional plants; and providing conditions to encourage children to engage in more physical activity. The PIP analysis was useful for understanding realistic expectations for the potential of BeHealthy and its inputs. Based on limited data, the authors concluded that the BeHealthy Program goals are achievable. No clear impacts measured, limitations in measurements also depending on data quality and consistency. |
| | BeHealthy Charities Aid Foundation Program | Analysis of Critical Quality Control Points (CCPs) within Program Impact Pathways (PIP) analysis. | | | |

*(Continued)*

**Table 1.** (Continued)

| First author/year | Platform/Programme | Country/Study design | Sampling | Intervention description | Outcomes |
|---|---|---|---|---|---|
| Paes-Sousa/ FF-IDS (2014–2017) | Community<br><br>Zero Hunger/Programa Academia Saúde: Brazil's food, nutrition and health promotion governance plan | Brazil<br><br>Not applicable (multi-year evaluations on multiple components which were designed and implemented at different points in time). | Not applicable—Brazil's school feeding program, the second largest in the world feeds 42 million of the country's school children. Mandated by government that 30 percent of the ingredients for school meals be sourced from local, family farms, involving around 4 million of the country's small farmers. | Comprehensive multisectoral efforts including: (1) incorporation of obesity in health, nutrition, food, and nutritional security policies, (2) mobilizing different governmental departments to conduct food and nutrition education, (3) building an integrated interdepartmental governmental response to prevent and control obesity, (4) promoting and providing healthy foods in schools, (5) directing healthy diet promotion and obesity prevention actions to occur in the primary health-care sector, (6) promoting physical activity in the community, (7) regulating the food industry advertising and marketing practices to young children, and (8) social action and community empowerment through mass communication and capacity development of local nongovernment organizations + over 200 human milk banks across Brazil. | Demonstrated impacts on nutrition and health outcomes, school attendance, improved nutrition security of children, improved awareness and knowledge of healthy foods in schools, increased opportunities for physical activity in schools and communities, increased awareness through advocacy campaigns, improved maternal health and IYCF practices, improved nutrition of disadvantaged infants through milk banks, increased livelihood security and income stability through smallholder farmers' involvement in food provision and through cash transfers, increased household food security, improved community knowledge and capacity, and community empowerment outcomes. |
| Patriarca/RIDOH (2016–2018) | Community<br><br>Rhode Island's Health Equity Zones | USA<br><br>Assessment of the Rhode Island Department of Health 'Health Equity Zone' Initiative | Not applicable. Target population is different for each HEZ collaborative, and design and implementation are planned by each working group based on local needs assessment. | Designed to address the many social and environmental factors contributing to the health–and unhealthiness–of a community, and differences between communities. A HEZ is an economically disadvantaged, geographically defined area with documented health risks. A group of volunteer stakeholders, organized as a "HEZ collaborative," works to achieve health equity for the residents of the HEZ by eliminating health disparities, and using place-based (where you live) strategies to promote healthy communities. A HEZ may be as small as several city blocks, or as large as a county. The size and boundaries of a HEZ are defined by stakeholders. | Mixed results. Each working group focuses on cluster of planned projects under categories such as food and nutrition, physical activity, substance abuse awareness and prevention, personal mental health and wellness, and other locally relevant components. Findings suggest that while a fully formed coalition may become self-sustaining (e.g. with appropriate resources for personnel and overhead costs such as rent, communication expenses, travel, etc.), a forming coalition usually requires several years' 'seed' funding to get it to the point where members are willing to invest in the enterprise. |
| Payne (2018) | Workplace<br><br>Workplace culture of health (COH) | USA<br><br>2013 and 2015 survey data from the National Healthy Worksite Program, a Centers for Disease Control and Prevention (CDC)-led initiative to help workplaces implement health-promoting interventions. | 41 employers completed the CDC Worksite Health Scorecard to document organisational changes. 825 employees provided data to evaluate changes in their health and attitudes. | Worksite-based health promotion programs (WHPPs) to reduce rising health-care costs, attract and retain talent, and improve employees' quality of life. Workplace Culture of Health (COH) as environmental, policy, and programmatic supports; leadership and co-worker support; employee engagement (motivational interventions); and strategic communication. | Relational elements of COH (leadership and co-worker support) tend to be associated with perceived support for health, while workplace elements (environmental and policy supports) are more often associated with lifestyle risk. Different degrees of integration of these elements. Findings suggest that employers need to confront relational and workplace elements together to build a COH. |

(*Continued*)

**Table 1.** (Continued)

| First author/year | Platform/Programme | Country/Study design | Sampling | Intervention description | Outcomes |
|---|---|---|---|---|---|
| Perez-Escamilla (2018) | School<br><br>Mondelez International Foundation Healthy Lifestyles Programs | China, India, Brazil, South Africa, UK, Germany, Mexico<br><br>A qualitative analysis was conducted to identify key enabling factors and map them into guiding principles for effective PPPs. | Triaged qualitative data from (1) proceedings from 2 school-based healthy lifestyles program evaluation workshops in October 2013 and in May 2016; (2) Mondelez International Foundation (MIF) annual country reports and MIF project reports; and (3) interviews with key program leaders from each program. | Innovative healthy lifestyles school-based Public-Private Partnerships designed to curb the childhood obesity epidemic globally. | Clear multisectoral governance structure was documented in 4 of the 7 countries. Consistent with the inclusion criteria and technical support offered by MIF, all countries had a clear baseline to monitor progress and measure success. Strong objective evidence from 3 programs (Brazil, Germany, UK) of special efforts to ensure key stakeholders were involved in decision-making process. 3 countries had strong evidence that flexibility regarding program implementation is an intrinsic component of their PPPs. The same 3 countries also engaged families with the school, through program elements that were formally incorporated into their PPP programs. |
| Reeve (2015) | Community<br><br>State and Municipal Innovations in Obesity Policy | USA<br><br>Review of local innovations in obesity policy | Not applicable. Wide variety of designs, implementation and evaluation plans across settings within holistic enabling environment approach. | Large number and wide variety of initiatives to improve healthy living. As well as standard food and nutrition initiatives at state and municipal level, examples of Innovation with a physical activity component include the following: Joint-use agreement: City and Tucson Unified School District opened up 12 school sites to the public, with police department providing extra patrols around sites for community safety. / Atlanta Beltline: ambitious project fusing economic development with transportation planning to create new small businesses, housing, parks, and transit along 22 miles of repurposed trails. / Play Streets, multicity initiative: closing streets to traffic and opening up reclaimed space for children's play and physical activity. / Druid Hills Revitalization Project, Charlotte: partnership between Charlotte Police and a local non-profit to improve safety in high-crime neighbourhoods, creating community taskforce, remodelling old housing stock, and raising funds for community park. / Sarah Vaughn Field of Dreams baseball park, Greenville; renovations to make spaces fully accessible to children with disabilities. | Evidence of a wide range of innovative strategies to tackle obesity from state and municipal level. Evidence of increased opportunities for healthy nutrition and physical activity framed as expected potential benefits of enabling environment and decreased exposure to risk factors, rather than measured impacts on health outcomes. |

(*Continued*)

**Table 1.** (Continued)

| First author/year | Platform/Programme | Country/Study design | Sampling | Intervention description | Outcomes |
|---|---|---|---|---|---|
| Rito (2013) | Community | Portugal | Five Portuguese municipalities and local communities. Children were not randomised to intervention and control conditions. 266 overweight children (BMI ≥ 85th percentile) aged 6–10 years, from low-income families in five Portuguese municipalities, were assigned to the intervention. | Program Obesity Zero (POZ), a multi-component, community-, family- and school-based childhood obesity intervention. Parents and children attended four individual nutrition and physical activity counselling sessions, a one-day healthy cooking workshop and two school extracurricular sessions of nutrition education. | Reported initial non-acceptability of participating in POZ from families with overweight children, which resulted in 55.1% less attendance. Measurements of BMI-for-age, waist circumference, intake of fruit and vegetables, fibre consumption, sugary drinks intake, physical activity levels and screen time suggested that POZ is a promising intervention, at municipality level, to tackle childhood overweight and obesity. Lack of longer-term follow-up and absence of control group limit reliability of findings. |
| | Program Obesity Zero (POZ), targeting children from five municipalities | Evaluation through statistical analysis of data. | | | |
| Sekhobo (2012) | Preschool | USA | All preschool-aged children enrolled in New York State Special Supplemental Nutrition Program for Women, Infants and Children (WIC). | Beginning in 2005, a state-wide childhood obesity prevention initiative known as NY Fit WIC was implemented to target preschool-aged children enrolled in the New York State (NYS) Special Supplemental Nutrition Program for Women, Infants and Children (WIC). The initiative used a train-the-trainer model. The training focused on learning age-appropriate movements and the use of simple inexpensive toys to support a life-long habit of being physically active. | Behavioural changes in participants, staff and caregivers. Increased activities or events implemented in the months following the training. Implementation of activities for (i) participants/caregivers education, (ii) support for WIC clinic staff, (iii) clinic activities promoting healthy lifestyle and (iv) community efforts. Beyond increased opportunities, findings on impact on meaningful behavioural outcomes among WIC staff, caregivers and children are found to be inconclusive. |
| | New York State childhood obesity prevention program for Women, Infants and Children (NY Fit WIC) | Evaluation through statistical analysis of data but programme did not have a control group. | | | |
| Soler (2014) | Community | USA | Not applicable. Multiple implementation and evaluation activities conducted as part of the Centres for Disease Control and Prevention's (CDC's) Communities Putting Prevention to Work (CPPW) initiative. | CPPW was one of the largest federal investments ever to combat chronic diseases in the United States. CPPW aimed to improve health for the largest number of people possible by implementing changes in policies, systems, and environments, and by setting priorities at the community and population levels on the basis of the principle that individual education and tertiary health care are insufficient to prevent chronic diseases. | Complex national programme. Overall changes in environments, decreased exposure to risk factors, increased opportunities for healthy lives, improved data collection and analysis for program planning, enhanced evaluation process for program improvement in rural childcare settings. Outputs included analyses of joint-use agreements and trail use, analyses of student consumption of school meals after changes in nutrition standards, educational media campaign about sugar content in beverages and dissemination of findings. |
| | The CDC's Communities Putting Prevention to Work (CPPW) initiative | Summary review of articles reporting on implementation and evaluation activities, to guide the evidence base for public health interventions on the basis of jurisdiction-wide policy and environmental-level improvements. | | | |
| SPRING (2014) | Community | Mexico | Population level. Prospera components targeted all age groups. The three components of the EsIAN strategy are (a) supplementation for pregnant and lactating women (tablets) and children aged 6–59 months (micronutrient powder, fortified porridge and milk), (b) improved health systems (specifically, equipment and quality of nutrition counselling), and (c) behaviour change communication and training. | Prospera is a national conditional cash transfer program that aims to improve the utilisation of public services by low-income households in Mexico across the health, education, and social development sectors. EsIAN is a national strategy to strengthen the health and nutritional component of Prospera by addressing undernutrition and obesity, with a focus on the first 1,000 days of life. | Multi-year evaluations of numerous components. Prospera/EsIAN reached 6.1 million families, or 26 million people, and trained more than 191,000 health workers. |
| | Prospera/EsIAN: Mexico's National Integrated Nutritional Strategy | Pilot, then multi-year evaluations on multiple components which were designed and implemented at different points in time. | | | |

(*Continued*)

**Table 1.** (Continued)

| First author/year | Platform/Programme | Country/Study design | Sampling | Intervention description | Outcomes |
|---|---|---|---|---|---|
| TYF (2014) | Community<br><br>The Health Innovation Network (HIN) and Young Foundation (TYF) Obesity South London Strategy | UK<br><br>Overview reporting on the effectiveness of several different interventions to tackle obesity in South London. | Not applicable. Includes multiple RCT studies, population level overview. | The HIN commissioned TYF to help with the development of an Obesity Strategy for South London, moving away from a health service focused primarily on treatment of ill health, to one more focused on prevention with public and local communities taking more responsibility for their own health and wellbeing, and better managing their own conditions. Focus on adults whose weight is a problem for them in their lives, but not yet so severely obese that surgery is the best option. Two main groups: those needing motivational support to prevent them tipping over into becoming overweight, and those who need specialist weight management to prevent them tipping over into becoming morbidly overweight where health problems escalate rapidly. | Increased opportunities for healthier lifestyles through implementation of multiple components including: self–monitoring, peer support, cognitive restructuring, stimulus control, education and advice, access to physical activity, access to dietary advice. Increased motivation, often via groups or gamification, including 'rewards' such as points and levels of achievement to 'nudge' people to change. App-based monitoring (e.g. MyFitnessPal), drinking monitoring, physiological feedback, e.g.Optima Life using Firstbeat heart monitoring. Effective development of partnerships which involved citizen representatives, innovators, energy retailers, supermarkets, restaurants and shops, exercise experts, implementers, local networks and boroughs, marketing experts. Improved monitoring and knowledge sharing through building and maintenance of data sharing platform. |
| Weber (2017) | School<br><br>Promoting Physical Activity and Balanced Diets in school with high proportion of children with a migration background | USA<br><br>Differences in changes using linear regression analyses. | Four 3rd and 4th grade classes (n = 70 children, 77% with migration background) participated in a 10-months intervention comprising 2 additional exercise lessons weekly and 10 nutrition lessons per school year. 6 school classes (n = 125 children, 65% with migration background) served as control. In a subgroup (n = 37), after 6 months of the intervention, daily physical activity was assessed by accelerometer-based monitoring. | Promoting Physical Activity and Balanced Diets in a Primary School Setting with a high proportion of children with migration background, in order to target groups at higher risk of obesity (e.g. age-group, children from migrant families). | Dietary behaviour and knowledge improvements. Increased physical activity. Findings show that promoting guided physical activity in a primary school setting with a high proportion of children with migration background positively affected parameters of fitness and motor skills. |

Visual syntheses of results relevant to review objectives were produced for ease of access. Given that the wide array of study types and differential data quality do not allow for statistical synthesis of the results, the distribution of evidence was plotted as a simple bar graph to display relative magnitudes without implying functional relationships. Graphs were used to represent the distribution of studies across delivery platforms, outcome measures reported across programmes relevant to unexploited pathways specified in the conceptual framework, and characteristics of programmes that are relevant to the assessment of double duty action potential.

Due to the inclusion of outcome metrics of differing rigour and potential for bias across the included study designs, this approach to visual synthesis presents a number of advantages: a) it enables to visualise the characteristics of programmes which are most relevant to the stated objectives of the scoping review, despite the non-comparability of data across studies; and b) it permits an agnostic position with relation to the outcomes and metrics used in the studies, in line with the decision to scope the literature including diverse methodologies.

This outline of the contribution of the evidence served as a first step in assessing whether and to what extent the double-duty potential of actions through planned or potentially unexploited impact pathways could be assessed, at a time when the coexistence of multiple burdens of malnutrition is increasingly recognised as an urgent issue across countries. At the same time, the graphs make it clear that the current evidence base would not allow for a thorough systematic review and meta-analysis at this stage. This is particularly important for the identification of gaps which will need to be addressed if we are to make progress on double duty action to tackle malnutrition in all its forms.

Double-duty potential was assessed through a further stage of analysis. Evidence of awareness and/or applied knowledge of the links between interventions and multiple drivers of different forms of malnutrition was sought and recorded (S2 Table). We assessed whether interventions explicitly addressed different forms of vulnerability and inequality including those relating to age, gender, socio-economic or ethnic status and cultural identity. Reported outcomes in relation to underlying and structural determinants were also assessed with regard to their potential for creating an enabling environment for DDAs, importantly for both '*denovo*' actions and '*retrofitting*' of existing single-duty actions [6]. The level of detail and focus of the studies included in this review, however, did not allow for an assessment of whether interventions framed as obesity prevention or control are consistent with the '*do-no-harm*' principle for double-duty action, whereby interventions addressing a single form of malnutrition ensure no harmful impact on other forms of malnutrition.

## 3. Results

The approach to study selection is shown in Fig 2. We identified a total of 675 records, including 620 from peer-reviewed database searches and 55 from grey literature searches. After removing duplicates (n = 206), a total of 359 peer-reviewed records were screened by title and abstract (55 grey literature articles, which did not provide an abstract, were screened in the following step). 98 articles, including 43 peer-reviewed and 55 grey literature articles were screened by full text. 31 peer-reviewed studies and 5 grey literature articles met our eligibility criteria.

In order to comprehensively assess the quality of studies and interventions, the WHO 'Good Practice Critical Appraisal Tool for Obesity Prevention Programmes, Projects, Initiatives and Interventions' [13] was adapted to the purposes of this review (S1 Table). A total of 36 studies satisfied quality assessment criteria and were included in this review.

Our results highlight a variety of different intervention types that meet inclusion criteria for this review. Reporting of the findings draws on information collected in the extraction matrix, detailed in the previous section. This is organised into six overlapping categories (Fig 3), which highlight entry points for implementation across the lifespan (the number of key studies that contributed to each category is specified in brackets): early care/preschool platforms (n = 7), school-based platforms (n = 14), out-of-school time (OST) platforms (n = 8), targeting of higher risk groups (n = 6), workplace platforms (n = 4), community-based initiatives (n = 24).

### 3.1. Early care/preschool platforms

Licensed child care centres and pre-kindergarten programmes provide an opportunity to reach large numbers of children, including those at risk of both undernutrition and overweight/obesity [32]. Factors considered in obesity prevention interventions include maternal health, nutritional and health literacy, family perceptions of healthy infant growth, family eating, cooking and exercise behaviour, and the role of obesogenic food environments [33, 34]. Traditional initiatives provide education for staff, food service training, healthcare provider

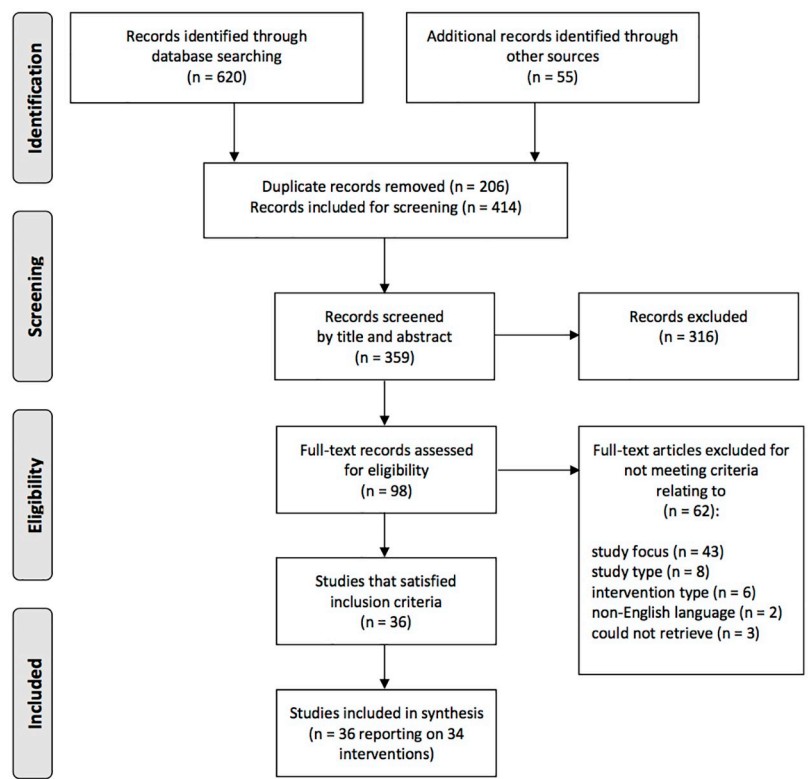

**Fig 2. Study selection flow chart.**

support, and target the attitudes of parents in relation to breastfeeding, milk banks, nutrition, and active lifestyle for their children. Some innovations include provider-child interactions around food and physical activity; emotion-based approaches aimed at parents' perceptions; learning of age-appropriate movements and use of simple inexpensive toys to support a life-

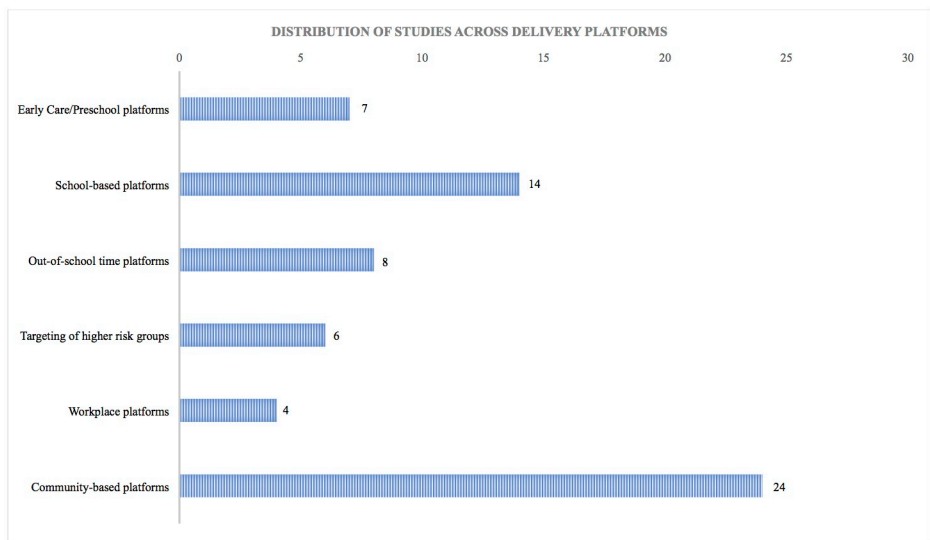

**Fig 3. Distribution of studies across delivery platforms.**

long habit of being physically active; opportunities for family-time in outdoor settings and active participation in community gardens [32–38]. Examples of innovation are evidence-based interventions which integrate new knowledge of developmental phenomena within the curriculum—e.g. the need for 8 to 12 exposures to a novel food before developing a preference for that food [37], or recommendations that preschool age children should have at least 60 minutes of unstructured (free-play) and at least 60 minutes of structured (adult-led) physical activity each day [37]. Although potentially effective in encouraging meaningful behavioural outcomes among caregivers and children, evaluations have often produced inconclusive results.

Several papers claim to have made some degree of progress in establishing the conditions for policy, systemic, and environmental changes. Examples of action include establishing multisectoral commitment to common agenda and priority-setting activities, asset mapping, provision of training and technical assistance, establishment of continuous communication networks and file sharing systems, shared quality rating methods, creation and alignment of resources that support childcare centres in family engagement strategies around breastfeeding, physical activity, and nutrition, drafting recommendations for improving existing programme wellness, nutrition and physical activity criteria, breastfeeding licensing and regulation standards for childcare programmes, and inclusion of nutrition and physical activity criteria in state quality rating improvement system [32, 36, 38]. However, specific outcomes are not measured or specified.

Evaluations of care centre-based initiatives often rely on self-reported data, and a few indicators that quantify the opportunities provided for improved healthy eating and physical activity [32, 36, 38]. Wellness policies and programmes focus on training caregivers in best practices for physical activity and nutrition that promote healthy weight for young children [38], healthy preschool time [32], development of a healthy environment conducive to healthy lifestyle choices [32], dietary behaviour at breakfast or over the course of one day [36], improving the physical activity environments for physical activity training and education and for active play, as well as the healthy weight environment by influencing the quality of foods served, caregivers' behaviour, and staff behaviour [32, 36]. Often due to imperfect data collection systems, they fail to demonstrate whether additional opportunities produce behavioural changes and desired health outcomes.

## 3.2. School-based platforms

There is evidence that schools can shift dietary behaviour and physical activity practices over the course of an academic year [39, 40]. Many school-based health promotion interventions have traditionally focused on changing individual behaviour, rather than targeting broader social or environmental determinants that influence behaviour. The promotion of health-conducive environments in schools is the focus of new interventions which provide opportunities for engaging in healthy nutrition and physical activity throughout the day [34, 40–43]. Innovations include attention to breakfast habits, food service in canteens and availability of snacks and drinks in vending machines, bans on drinks except milk and water, healthy snacking breaks, provision of free fresh fruit in the classroom, and engagement with food retailers in the school's surroundings [36, 37, 43–46]. Physical fitness and motor skills tests, awareness building and questionnaires on dietary habits are administered for monitoring uptake of healthy behaviours [47]. Some initiatives reinforce the communication of health-related messages by employing age-appropriate educational materials informed by dietary guidelines, online learning, and messaging [34, 48]. Limitations have been found in the use of generic materials provided for educational purposes that do not consider local contexts, e.g. availability and

seasonality of fruit and vegetables, or tweaking of locally-relevant culinary styles and methods [37].

Multicomponent interventions in schools that combine educational, curricular, and environmental elements are thought to be more effective than interventions targeting single components or behaviours [39]. High-intensity interventions that focus on multiple aspects, interventions with periods of time greater than six months in duration, sustained funding, and parental involvement in the content and/or planned actions, are frequent recommendations for achieving effectiveness of programmes [36, 43–45, 47, 49]. Holistic approaches that promote a supportive 'school ethos' and emphasize improvements in physical, social, and emotional well-being and educational outcomes refer to the children's health-related quality of life (HRQoL) measured using the Health Utilities Index (HUI) [44]. Others measure students' quality of life in school with the Quality of Life in School (QoLS) instrument [39]. This is a measure of general well-being and satisfaction based on positive and negative experiences of school activities which target diet quality (defined by adequacy, variety, balance and moderation), physical activity, screen time, and self-efficacy. The findings suggest the potential role of a supportive school ethos for well-being in school. But further, longer term and longitudinal research is needed to demonstrate the potential effects of such approaches on student health, well-being, and academic achievement into the future [39, 44].

## 3.3. Out-of-school time (OST) platforms

Some interventions focus on out-of-school time (OST) programmes as key environments for targeting school-aged children and adolescents. Dissemination of guiding principles for healthy snacks, beverages, and physical activity in large and complex OST organisations has been observed to be most successful when implementation strategies are customised, aligned and integrated with existing goals and routine practices [48, 50]. Competing demands for time in the curriculum have been found to be a barrier to participation [50, 51].

One major difference between school-based and afterschool initiatives appears to be a shift from a view of beneficiaries as passive recipients of interventions to one of active participants. A novel approach in programmes that combine nutrition and physical activity, is the engagement of young people in advocacy activities. Initiatives focused on youth engagement in public–private partnerships, record impacts on policy and industry behaviour [41, 52]. The use of 'star power' of professional sport organisations with celebrity status in collaboration with youth groups has shown positive outcomes at the individual, group and community levels [45, 50]. By getting directly involved, young people become more aware of their own behaviour and take steps to get healthier. Additionally, they exert influence on other young people as well as their families. Interventions that impact on health and wellbeing of adolescents (e.g. through increased opportunities for healthy eating, and increased uptake of moderate to vigorous physical activity (MVPA) components) offer second chances to the most disadvantaged [43, 50, 51, 53]. Investments in adolescent health and wellbeing bring benefits not only in their future adult life, but also for the next generation of children as future parents.

Components which focus on the promotion of health and wellbeing of children and adolescents include the integration of evidence-based, practical, and accessible guiding principles for promoting drinking water instead of sugar-sweetened beverages, boosting movement and physical activity in all programmes, and fuelling up on fruits and vegetables [52]. There is evidence of increased moderate or vigorous physical activity, replacement of flavoured milk with non-fat or regular milk, modest improvements in mean numbers of healthy food items served per day, increase in awareness of healthy labelling through promotional materials and healthy food labels but no significant increase in purchase of fruit and vegetables or other healthy

items [43], increase in caregivers' nutrition-related efficacy beliefs, increased knowledge on healthy diets and physical activity in children and caregivers [51]. Whilst a definition or exact measure of wellbeing is not specified in these studies, the development of actionable guiding principles for healthy eating and physical activity in OST platforms is often combined with an existing portfolio of national initiatives that blends science and business strengths to prevent childhood obesity, with general health and wellbeing as broad outcomes [41, 43, 45, 50, 52, 53].

## 3.4. Targeting higher risk groups

Some interventions intentionally target school districts with a high percentage (at least 50%) of students eligible for free or reduced-price meals (FRPM), from urban, suburban and rural regions, for increased likelihood of impact [49]. Others focus on schools with a high percentage of students from families considered to be at higher risk of obesity in some settings, e.g. from migrant minorities [47, 51]. In addition, multidisciplinary, family-centred outpatient interventions based on social cognitive theory have been implemented to prevent further increases in BMI and improve quality of life (QoL) in children and adolescents with obesity in healthcare-based settings and through home-visits [46]. Though recent data suggest that these can have a positive effect on childhood obesity, with impacts on anthropometric and laboratory value outcomes, it is still unclear which program components are most beneficial and how they affect QoL, or whether home visits contribute to stigma [34, 39, 46]. The evidence shows mixed results. Some initiatives to promote healthy weights in children from vulnerable populations report no changes in measurements of health-related quality of life in children (HRQoL), based on previous month recall questionnaires measured at baseline, endline, and six-month follow-up, and no differences in health-related quality of life within sites covered by these initiatives [51]. One study found that following a 1-year intervention, the participants' BMI z-scores and QoL improved, while other adiposity-related measures of body composition remained unchanged [46]. These studies stress the need to generate more data on health-related QoL outcomes [46, 51].

## 3.5. Workplace platforms

In recent decades, employers have increasingly established workplace-based health promotion programmes to reduce rising health-care costs, attract and retain talent, improve employees' quality of life and minimise absenteeism [54, 55]. The intermediate aim is to help employees adopt healthier lifestyles and lower their risk of developing costly chronic diseases, while improving worker productivity. Some initiatives have focused on health promoting workplaces (HPWPs) as ideal settings for influencing the physical, mental, economic and social wellbeing of workers (and in turn their families, communities and societies), creating large multisectoral networks that engage in training of trainers, toolkit development for employers and workers and promotion of healthier food and physical activity environments, as well as decreased exposure to other risk factors for non-communicable diseases and injuries (NCDIs) [56].

Programmes typically entail an assessment of employee health, personalised feedback on how employees can improve their health, and the provision of resources and programming designed to promote wellness [55]. The stress is on creating an enabling environment for health promotion, addressing sedentary behaviours as a leading cause of NCDs in non-labour-intensive industries, promoting awareness and potentially fuelling public demand for health-conscious policies in other domains of adult life, as well as those aimed at prevention during childhood and adolescence. Scores of employees' perceptions of employer support for health

and lifestyle risk is derived from self-reported physical activity, nutrition, and other risk factors, such as tobacco use [55, 57]. Science- and practice-based prevention and wellness strategies are being implemented in different countries, with the expectation that these will lead to specific, measurable health outcomes to reduce chronic disease rates [54–57]. However, these programmes employ wellbeing as a broad guiding principle rather than as a specific impact measure. [Percentages in Fig 4 represent outcomes for which specific measures were used.]

## 3.6. Community-based initiatives

Complementary to initiatives aimed at high-risk segments of the population are interventions that aim to change policies, systems, and environments, including priority-setting at the community and population levels. This reflects an awareness that end-stage treatment of chronic disease is not sufficient, and that there is a need to promote healthy lifestyles across the lifespan, and to address risk factors in multiple settings [58], including more deep seated health inequities based on high levels of income inequality. Notable examples are large scale development interventions aimed at broad societal change, such as the 'Prospera/ESIAN' Programme in Mexico [59] or the 'Zero Hunger/Academia Saúde Program' in Brazil [60, 61]. Both are based on a systems-thinking approach. EsIAN, is a national strategy to strengthen the health and nutrition component of the previously implemented Prospera conditional cash transfer programme. This addresses undernutrition and obesity, with a focus on the first 1,000 days of life [59]. The Zero Hunger Program combines comprehensive multisectoral efforts that incorporate obesity in health, nutrition, food, and national security policies, including the promotion of physical activity in the community, the regulation of food industry advertising and marketing practices to young children, and social action and community empowerment through mass communication and capacity development of local nongovernment organisations [61].

Beyond large scale policy-led initiatives, there is contrasting evidence as to the reach, implementation, adoption, effectiveness and sustainability of programmes which tackle the obesogenic environment [39, 54, 62–64]. Existing evidence on the extent to which social and community-based approaches incentivise exercise and awareness is inconclusive [34, 62]. A major issue in assessing impact remains the quantity, quality and consistency of data collection

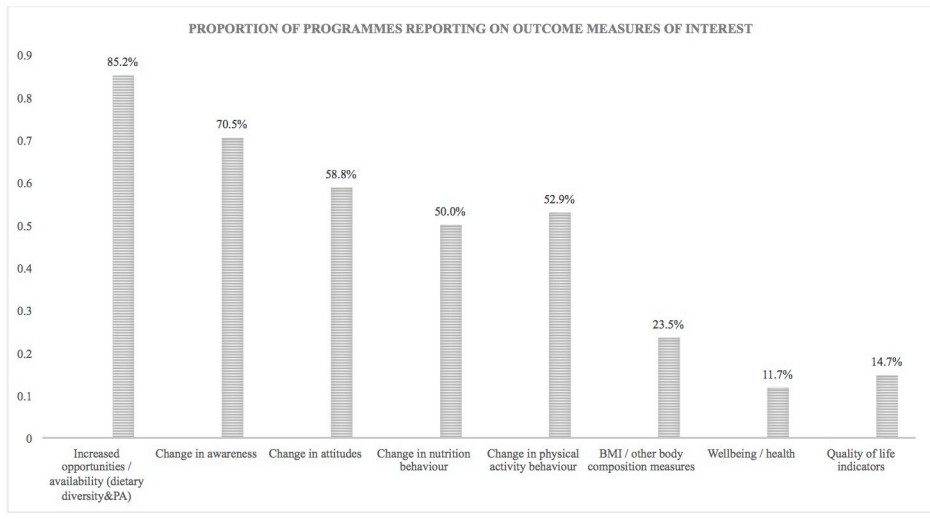

**Fig 4. Proportion of programmes reporting on outcome measures of interest.**

and monitoring. Nevertheless, there is evidence of positive change in awareness and behaviour that are likely to be conducive to positive outcomes in the prevention of multiple forms of malnutrition [41, 49, 65]. Interventions that encompass the multiple contexts that influence people's choices (i.e. family, school, work, community, culture and society) and related policy arenas, are believed to be crucial for the prevention of comorbidities in children, adolescents, and adults [48, 66]. Analysis of post-intervention data shows a broad range of impacts at micro-, meso- and macro-level, without allowing for a conclusive assessment of intervention components in absolute terms, stressing how the extent to which change is influenced by single or combined components is highly contextual.

Some initiatives use traditional intervention platforms but combine them into multicomponent, community-family-school-based childhood obesity interventions [51, 52, 67]. Intervention components include attendance of individual nutrition and physical activity counselling sessions for children and parents, healthy cooking workshops, and school-based extracurricular sessions of nutrition education. Measurements from similar programmes of indicators such as waist circumference, BMI-for-age, intake of fruit and vegetables, fibre consumption, sugary soft drinks intake, physical activity levels and screen time, suggest that multicomponent interventions are a promising strategy, at municipality level, to tackle childhood overweight and obesity [32, 39, 46, 52]. These studies record improved dietary quality in terms of dietary intake and eating patterns, diet adequacy, variety, balance and moderation, lower total energy, protein, carbohydrate, sugar and total fat consumption, decreased contribution of sugary beverages to total daily energy intake, replacement of sugar-sweetened beverages with more nutritious fluids, higher intake of milk products, higher intake of dietary fibre, and increased consumption of fruit and vegetables, with changes being sustained at six months from intervention [32, 39, 46, 52]. Some programmes also highlight major barriers to uptake and effectiveness. Importantly, initial resistance to the programme by families with overweight children, results in low attendance [34]. Some of the strengths are the standardised intervention protocol for consistent delivery across settings, and the establishment and promotion of a partnership between health centres and local governments. However, despite positive outcomes in the short term, the lack of longer-term follow-up makes these results less robust for drawing conclusions on sustained change within targeted communities [51, 52].

More effective, higher-intensity strategies are more likely to be found in communities with the longest duration of investment [34, 43, 56, 62]. Strong (high-dose) community-based obesity prevention strategies have been shown to lead to improved health behaviours, with increased opportunities for healthy eating and physical activity, and measured impacts on moderate to vigorous physical activity, dietary diversity, increased awareness achieved through educational materials and food labelling, decreased intake of sugar-sweetened beverages, and increased intake of fruits and vegetables [43, 49]. One programme, aimed primarily at children from vulnerable populations (i.e., rural, remote, northern, aboriginal and multicultural communities across Canada) which employed similar strategies, however, was found to have limited effectiveness [51], with healthier cooking and eating choices and play-based physical activity being reported during implementation of the programme, but with no long-term effectiveness. Importantly, motivation has been found to be stronger when beneficiaries are involved, supported and encouraged [43, 49, 53]. Comprehensive evaluations that systematically highlight areas of success and challenges are not available for most programmes, which present self-reported changes in awareness and behaviour but no objective measurement of their extent, making it difficult to compare interventions [38, 51].

There is positive evidence of programmes focused on components such as social marketing, stakeholder engagement, network and partnership development, community-directed needs

assessment and capacity building, showing changes in community capacity, but no or few impacts on individual knowledge, beliefs, perceptions and behaviour with relation to healthy eating, physical activity and sedentary lifestyle habits [34, 51, 54, 58, 62]. In contrast, some programmes—depending on their specific characteristics, including the number and duration of mutually reinforcing activities—show improvements in awareness, knowledge and behaviour indicators, but no measurable impacts at the community level, beyond the potential establishment of environmental conditions conducive to healthy behaviour [38, 51]. Studies which compare implementation, uptake and impact of the same initiatives or programmes across communities [54, 62] have generated divergent results with regard to a number of indicators. Projects are as diverse as the local problems they address (including substance abuse, diabetes, access to fresh foods, access to primary health care and mental health care, problems with the built environment, and equity issues within communities) [62]. These findings show that effectiveness of intervention strategies is dependent on individual and community factors.

## 4. Discussion

In this review, we sought to synthesise and summarise major initiatives that have integrated physical activity and nutrition promotion, with potential synergistic effects on multiple forms of malnutrition, in order to distil some key lessons for informing the future investigation, design and implementation of nutrition-relevant actions. We find that 33 of the 36 studies meeting inclusion criteria are limited to focusing or reporting on obesity outcomes. Also notable is the fact that 22 of the 34 programmes that satisfied quality assessment were implemented in one country (USA), with 3 others from Canada, and 1 each from an additional 8 countries, and just one multi-country initiative.

Only two interventions (reported by 3 of 36 studies) were explicitly targeted at *both* obesity/overweight and other forms of malnutrition. These are complex, large scale, multisectoral programmes in countries (Mexico and Brazil) with a high burden of both undernutrition and overweight/obesity, and related comorbidities. Their implementation of a wide range of components involves concerted action across many sectors and policy levels, with the specific objective to address multiple forms of malnutrition. Some of the components encapsulate years, in some cases decades, of social protection policy, while others are more recent additions.

The remaining programmes included in the review were considered to have potential for 'double duty' based on several considerations. Consistently with the Cochrane PROGRESS-Plus framework [7], we applied an equity lens to the classification of studies, based on their recorded awareness of/focus on characteristics that stratify health opportunities and outcomes (Fig 5). Demographic and socioeconomic factors associated with malnutrition were taken as important markers of potential for interventions to perform double duty (S1 Text). These elements encompass multiple vulnerabilities influenced by underlying and structural determinants of malnutrition, as well as recognised environmental exposures to risk throughout the lifecycle. The attention given to vulnerabilities and inequalities in the studies examined were deemed important for all forms of malnutrition. A stress on environmental aspects was therefore considered as having potential for reducing the differential effects of the PROGRESS-Plus factors, importantly for both '*retrofitting*' and '*de-novo*' actions, by targeting multiple drivers and facilitating broader changes that are conducive to double duty. Additional attention was paid to consistency with the '*do-no-harm*' principle. While this extra layer of analysis was precluded by a lack of sufficient information in reviewed studies, it will be crucial for strengthening the knowledge base for future action.

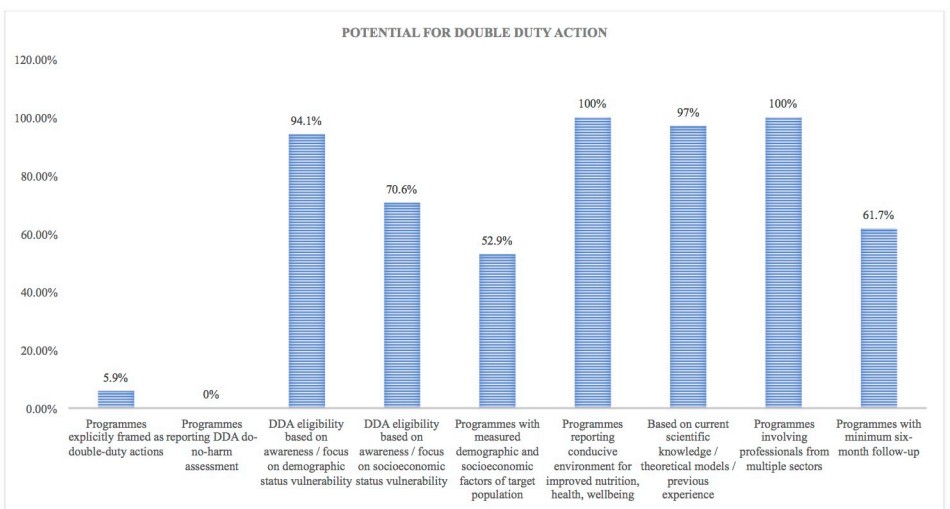

**Fig 5. Potential for double duty action.**

As illustrated in Fig 5 (and explained in detail in S1 Text on the rationale adopted), those studies demonstrating an awareness of/focus on known links between demographic and socio-economic factors, differences in endowments of individuals and groups, associated circumstances and behaviours adopted by individuals and households to cope with limitations in given settings, and consequent inequalities in health outcomes and life expectancy within and across communities, were considered to have potential for double duty.

Most health promotion interventions have traditionally focused on changing individual behaviour as an immediate determinant of ill health, rather than targeting broader socio-political or environmental determinants that influence behaviour [35, 39, 41]. Healthy nutrition and physical activity participation however are acknowledged to be dependent on factors beyond an individual's control, with programming and intervention efforts necessary across the lifespan, in multiple settings, and under various life circumstances [68]. To affect mass change, those involved in programming and delivering nutrition and physical activity interventions must address not only individual lifestyle behaviours that contribute to malnutrition and related comorbidities, but also the underlying and structural causes of health inequalities, perceptions of body and health, and stigma [69]. This shift in emphasis has created opportunities and challenges for those involved in nutrition and physical activity programme delivery. Instead of end-stage treatment of chronic disease in healthcare settings, the new focus on prevention across the lifespan entails a shift in resources towards promotion of healthy lifestyles, and early recognition and treatment of risk factors and symptoms in multiple settings [70], which is implicit in much of the literature.

Such broader considerations mean that multi-component and longer-term interventions are needed to ensure sustainability beyond the initial outcomes that may be measured by some of the shorter term, single component interventions. There may be a trade-off between the intensity of the intervention that can be delivered and resources needed to sustain intensive, multi-component, multisectoral programmes over time. But time-limited, intense interventions are not sufficient if a return to one's normal environment means a return to the range of individual and structural factors which might shape behaviour [64]. There are also potential savings from disinvesting in interventions which are unlikely to achieve long term or systemic results.

The evidence reviewed also suggests effective targeting needs to be taken into account in intervention design and implementation as well as informed decisions about the different needs and proclivities of different population groups. Nutrition interventions targeting mothers, infants and children, are known to have considerable benefits for the prevention of all forms of malnutrition. But this does not imply that we ignore the needs of other affected populations. Studies here report multiple benefits of targeting affected populations later in the life-course, including targeting of adolescents [45, 50, 53]; or targeting adults in the workplace for prevention and control of comorbidities later in life [55–57]. High risk groups can also be specifically targeted via referral services or community and home outreach [34, 39, 46, 49, 51].

Across the studies we also see the importance of understanding the wider context of implementation highlighted repeatedly. Programmes require tailoring to the social, economic, cultural, and demographic features of a region, with attention paid to identification of the best strategies in relation to age, gender, socioeconomic status, cultural identities and spheres of influence of the participants [51, 53, 58, 62]. Engagement with local stakeholders and place-based planning of strategies is key to sustained change [43, 54], ensuring relevance to people's day to day lives, and the identification of community-specific barriers to uptake and adherence. Studies often highlight the importance of strengthening the capacity of people and networks already active in the community and building on existing initiatives, instead of adding new ones [34, 62, 63].

In highlighting these findings, we acknowledge a number of limitations of this review and across the studies examined. Although we include an additional assessment of study quality, the findings reported above are drawn from a range of studies of variable quality, including from grey literature sources. We did not exclude such findings as we think they warrant further discussion and evaluation as part of future interventions and more rigorous research and evaluation design. More broadly, while studies reviewed report on impacts at micro-, meso-, and macro-level, their findings vary. Some report varying degrees of impact on individual awareness, knowledge and behaviour, but no impact at community level [46, 47], whilst others report impacts on underlying and structural determinants at environmental level, but no measured impact on individual awareness, knowledge and behaviour [38, 51, 54, 65]. There were some studies which show measured impacts in setting-specific outcomes but study participants reverting back to usual behaviour in other settings in which they conduct their everyday lives [34, 65]. Others show effects in the short term (during the implementation window) but no measured effect on longer term outcomes [39, 49, 51]. There were also some studies which demonstrated impacts on programme outcomes without affecting (or possibly exacerbating) health inequalities [42].

## 5. Conclusions

This review has addressed an important knowledge gap in the growing field of double duty research. By mapping major interventions and initiatives that combine nutrition and physical activity components, it has identified actions that hold potential for tackling not only obesity, but healthy diets, sedentary behaviour and quality of life more generally, as well as actions that explicitly tackle multiple forms of malnutrition. Importantly, it has identified crucial gaps in current methods and praxis that call for further practice-oriented research, in order to exploit the synergistic effects of integrated interventions on outcomes other than obesity. Future design and evaluation of multisectoral approaches will benefit from an explicit framing of interventions as double-duty oriented. At the very least, single issue interventions should attempt to specify whether and how they ensure that no harm is caused to other forms of malnutrition and wellbeing. Implementers and evaluators alike would benefit from a clearer

framework of how this is to be achieved, in order to satisfy the requirements for effective double duty action set out by the World Health Organization [6] and, more broadly, for the achievement of the Sustainable Development Goals (SDGs). This needs to be facilitated by those pushing for double duty action, so that evidence for independent evaluation can be made available through more conscious and comprehensive design.

## Supporting information

**S1 Table. Critical appraisal.** Adapted from WHO (2011) 'Good Practice Appraisal Tool for Obesity Prevention Programmes, Projects, Initiatives and Interventions'. Studies marked with * are to be considered in conjunction with other studies on the same interventions.
(PDF)

**S2 Table. Assessment of double duty action eligibility.** Legend: Y = Yes; N = No; N/A = unclear / lack of detail for in-depth analysis of single components; - = need for in-depth analysis of single components.
(PDF)

**S1 Text.**
(DOCX)

## Author Contributions

**Conceptualization:** Stuart Gillespie, Nicholas Nisbett.

**Formal analysis:** Laura Casu, Stuart Gillespie, Nicholas Nisbett.

**Methodology:** Laura Casu, Stuart Gillespie, Nicholas Nisbett.

**Writing – original draft:** Laura Casu.

**Writing – review & editing:** Laura Casu, Stuart Gillespie, Nicholas Nisbett.

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
