## [Decision Letter · Decision Letter 0]

3 Feb 2020

PONE-D-19-32917

Integrating nutrition and physical activity promotion: a scoping review

PLOS ONE

Dear Ms Casu,

Thank you for submitting your manuscript to PLOS ONE. After careful consideration, we feel that it has merit but does not fully meet PLOS ONE’s publication criteria as it currently stands. Therefore, we invite you to submit a revised version of the manuscript that addresses the points raised during the review process.

Please take into consideration that Reviewers have essential reservations about applied methods (e.g. search strategy) and the way the results are presented in your manuscript. In general, there is also a general feeling that it lacks clarity in the presentation of key concepts and results. 

We would appreciate receiving your revised manuscript by Mar 19 2020 11:59PM. To enhance the reproducibility of your results, we recommend that if applicable you deposit your laboratory protocols in protocols.io, where a protocol can be assigned its own identifier (DOI) such that it can be cited independently in the future. For instructions see: http://journals.plos.org/plosone/s/submission-guidelines#loc-laboratory-protocols

We look forward to receiving your revised manuscript.

Kind regards,

Mariusz Duplaga, Ph.D., M.D., Ass. Prof.

Academic Editor

PLOS ONE

Journal Requirements:

2. Thank you for your submission to PLOS ONE. We note that your literature search included studies published till October 2018;to allow an up-to-date view of the topic, we would request that the search is updated. Moreover, please clarify why studies from the Public Library of Science (PLoS) and The Lancet were searched for separately (lines 158-159).

3. Our internal editors have looked over your manuscript and determined that it is within the scope of our Determinants, Consequences and Management of Obesity  Call for Papers. This collection of papers is headed by a team of Guest Editors for PLOS ONE:Rachel Nugent and Pratibha V. Nerurkar. Additional information can be found on our announcement page: https://collections.plos.org/s/obesity-one.  

If you would like your manuscript to be considered for this collection, please let us know in your cover letter and we will ensure that your paper is treated as if you were responding to this call. If you would prefer to remove your manuscript from collection consideration, please specify this in the cover letter.

Reviewers' comments:

Reviewer's Responses to Questions

**Comments to the Author**

1. Is the manuscript technically sound, and do the data support the conclusions?

Reviewer #1: Yes

Reviewer #2: Partly

2. Has the statistical analysis been performed appropriately and rigorously? 

Reviewer #1: N/A

Reviewer #2: Yes

3. Have the authors made all data underlying the findings in their manuscript fully available?

Reviewer #1: Yes

Reviewer #2: Yes

4. Is the manuscript presented in an intelligible fashion and written in standard English?

Reviewer #1: Yes

Reviewer #2: No

5. Review Comments to the Author

Reviewer #1: Dear Editor,

Thank you for the opportunity to review the manuscript "Integrating nutrition and physical activity promotion: a scoping review" (PONE-D-19-32917) for you valuable Journal.

The aim of this scoping review is novel and very challenging. I appreciated the effort to perform grey literature search, the assessment of the quality of the studies adapting the WHO Critical Appraisal Tool and, in particular, the double duty and life-course approach used to frame this interesting topic.

The main issue coming from the review in its current form is the hard readability of results and the lack of synthesis in the presentation of key messages. Clarifications on methodology aspects such as the search strategy and the eligibility criteria are also needed.

Overall, this is a good quality scoping review that can be useful for implementation and assessment of future nutritional and physical activity interventions, supproting a more comprehensive perspective.

I have some major and minor comments reported below:

Methods:

Page 7, lines 160-169: The search strategy used by study authors is very generic, probably in order to be comprehensive according to review aims. Surprising is the relatively small number of overall retrieved studies by database search that appears to be lower than expected from a so generic search strategy. Could you provide the exact full search strategy, maybe in appendix, used for each database in order to make the review results more reliable?

Page 7, line 165: Why you choose the timespan 2010-2018 as publication period? Please justify.

Page 7, Methods section: the study designs eligible for inclusion are not reported in this section. Some of the included studies are review and overview of studies and this is uncommon for systematic and scoping reviews that generally use previous review to retrieve other original studies rather than including them as they are. If in this specific case you think this could be more appropriate, please specify why and what kind of papers you consider eligible more clearly.

Results:

Page 8, Results section: This is a general comment on the results reporting. In a scoping review it is expected to find results synthesized, framed and visually reported according to a conceptual framework helping to answer the research question. You correctly reported the framework used to describe included interventions. What I think is missing in this version is a “visual” representation of review results, maybe of the “key lessons” main points. This can be very helpful to readers in better understanding the contribution of included studies and of the review overall for future practice and research. Can you provide a more “user-friendly” reporting of the main results of your review? Please report the method you will apply in the Extraction and analysis section.

Discussion and Conclusions:

Page 22, Discussion and Conclusion sections: According to my previous comment, the key points coming from this review are reported narratively and not easy to be comprehensively understood. Even if the discussion paragraph helps to synthesised the main take home messages, an additional paragraph including a short message related to the implications for practice and/or implications for research summarizing the key concept could be useful. This can be done also taking part of the conclusions that are quite long and not straightforward. I suggest again to synthetize in Results section the key elements pointed out by included studies according to your focus and framework, and to firmly refer to this “visual” synthesis to support your statements in the discussion and conclusions sections.

Minor comments:

Abstract – the abstract Methods and Findings section should be extended (maybe reducing the background and splitting this in two sections, methods and findings) to give more importance to key messages highlighted by the review.

Reviewer #2: This study question is of public health interest given the high prevalence of obesity in youth, adolescents and adults around the world and this paper focuses on interventions including a nutrition and physical activity component to reduce obesity.

Abstract:

The first paragraph is very confusing.

1. Introduction:

The first couple of paragraphs are quite convoluted and it is difficult for the reader to determine the direction of the authors' message and purpose.

DDA's are not well explained, not sure of their importance as it relates to the purpose of the review.

The introduction is verbose and quite exhausting to read. It would be beneficial to the reader if the introduction was reduced to 3-4 paragraphs and linearly constructed to lead the reader from the problem to the purpose of the study.

In the scientific literature, many reviews have been conducted on the topic of nutrition and physical activity interventions. How is this review different? What does this review contribute to the literature that other studies have not presented?

Methods:

From what I can tell, the readers did not provide how/if methodological quality was assessed for these studies. How many of theses studies were classified as high quality? moderate quality? low quality? This assessment provides perspective on the level of evidence for the association between nutrition/PA and obesity-related outcomes.

Also, while the outcomes are provided, the reader is not informed on the assessment of these outcomes, which are critical to evaluating the level of evidence. Also, more details on the interventions is suggested. Ex: The dose of PA? For 5-year interventions, did the intervention exposure change over time? Were there dissemination aspects to these interventions? It would also be interesting to comments on the participant burden and cost-to-benefit ratio, however that may be beyond the scope of the review or simply added to the discussion would suffice.

Results:

In most reviews, it is common for authors to provide numeric data. Some is provided here, but what percent were successful? Unsuccessful? The different platforms was an informative layout. I think added more quantitative information would also be helpful.

6. PLOS authors have the option to publish the peer review history of their article (what does this mean?). If published, this will include your full peer review and any attached files.

Reviewer #1: No

Reviewer #2: No

---

## [Author Response · Author response to Decision Letter 0]

18 Mar 2020

REVIEWER 1

Dear Editor,

Thank you for the opportunity to review the manuscript "Integrating nutrition and physical activity promotion: a scoping review" (PONE-D-19-32917) for you valuable Journal.

Response: We are very grateful to both reviewers for their commentary, which we feel has helped us improve the paper substantially. We address each comment below in turn.

GENERAL COMMENTS

1. The aim of this scoping review is novel and very challenging. I appreciated the effort to perform grey literature search, the assessment of the quality of the studies adapting the WHO Critical Appraisal Tool and, in particular, the double duty and life-course approach used to frame this interesting topic.

2. The main issue coming from the review in its current form is the hard readability of results and the lack of synthesis in the presentation of key messages. Clarifications on methodology aspects such as the search strategy and the eligibility criteria are also needed.

Response: These points have been addressed, see more detailed comments below.

3. Overall, this is a good quality scoping review that can be useful for implementation and assessment of future nutritional and physical activity interventions, supporting a more comprehensive perspective.

I have some major and minor comments reported below:

DETAILED COMMENTS: 

Methods:

4. Page 7, lines 160-169: The search strategy used by study authors is very generic, probably in order to be comprehensive according to review aims. Surprising is the relatively small number of overall retrieved studies by database search that appears to be lower than expected from a so generic search strategy. Could you provide the exact full search strategy, maybe in appendix, used for each database in order to make the review results more reliable?

Response: Done. Details of the search strategy have been clarified in the main text. A double duty focus makes the search syntax more specific to the review’s objectives still allowing to capture programmes that were not framed as double duty actions. After initial searches during syntax building, this was preferred to a broader focus which was found to retrieve a very large number of irrelevant results. The search was then updated to cover the time up to 31 January 2020 as requested by the editors. The search rationale and strategy for the review update was specified above and results were incorporated in the PRISMA diagram and manuscript. 

5. Page 7, line 165: Why you choose the timespan 2010-2018 as publication period? Please justify.

Response: 2010 coincided with the publication of WHO Global Recommendations on Physical Activity for Health, and of the Global Status Report on Noncommunicable Diseases 2010, the first report on the worldwide epidemic of cardiovascular diseases, cancer, diabetes and chronic respiratory diseases, along with their risk factors and determinants. The review has been updated to cover the decade to date. 

6. Page 7, Methods section: the study designs eligible for inclusion are not reported in this section. Some of the included studies are review and overview of studies and this is uncommon for systematic and scoping reviews that generally use previous review to retrieve other original studies rather than including them as they are. If in this specific case you think this could be more appropriate, please specify why and what kind of papers you consider eligible more clearly.

Response: As the topic is not yet mature enough for a systematic review, we did not restrict our search to specific study designs, in line with guidance for conducting rigorous scoping reviews (Arksey & O'Malley 2005; Pham et al. 2014). The approach suggested by the reviewer would have indeed been appropriate if the included reviews had been systematic or literature reviews. However, in this case, the six summary reviews/overviews included are reviews of programmes implemented within the same jurisdictional localities and/or under the same frameworks to tackle malnutrition and related comorbidities. These include multi-tiered programmes implemented at national level (e.g. Brazil, Mongolia, USA) or other jurisdictional level (e.g. South London, USA subnational level) and refer to initiatives with an underlying common objective or framework, similarly to multi-country evaluations of the same programme. Differently from the inclusion of literature or systematic reviews of unrelated initiatives, summary reviews of multi-component/multi-locality programmes were deemed crucial to the scoping of the evidence base. Especially given the aim to identify double-duty-oriented actions, the inclusion of these summary reviews was considered fundamental. To further reinforce this point, the Brazil case was one of only 2 programmes over a total of 34 which were explicitly framed as double-duty action. Its exclusion would have made the reporting of the findings inaccurate. Additionally, in view of the novelty of double-duty action policy recommendations, which include not only ‘de-novo’ actions, but also double-duty-oriented ‘do-no-harm’ analysis and ‘retrofitting’ of existing single-duty interventions, the exclusion of summary reviews/overviews of complex programmes would weaken this and future reviews. 

Results:

7. Page 8, Results section: This is a general comment on the results reporting. In a scoping review it is expected to find results synthesized, framed and visually reported according to a conceptual framework helping to answer the research question. You correctly reported the framework used to describe included interventions. What I think is missing in this version is a “visual” representation of review results, maybe of the “key lessons” main points. This can be very helpful to readers in better understanding the contribution of included studies and of the review overall for future practice and research. Can you provide a more “user-friendly” reporting of the main results of your review? Please report the method you will apply in the Extraction and analysis section.

Response: Done. A visual synthesis of the information that could be aggregated was provided through the addition of three graphs (Figures 3,4,5), which were added to the Results section. The rationale behind our choice was specified in the Extraction and Analysis section. Any other relevant text was added in proximity of the added graphs. 

Discussion and Conclusions:

8. Page 22, Discussion and Conclusion sections: According to my previous comment, the key points coming from this review are reported narratively and not easy to be comprehensively understood. Even if the discussion paragraph helps to synthesised the main take home messages, an additional paragraph including a short message related to the implications for practice and/or implications for research summarizing the key concept could be useful. This can be done also taking part of the conclusions that are quite long and not straightforward. I suggest again to synthetize in Results section the key elements pointed out by included studies according to your focus and framework, and to firmly refer to this “visual” synthesis to support your statements in the discussion and conclusions sections.

Response: Done. Figures 3, 4 and 5 have been incorporated in the results and discussion sections. We trust that the consistency of the narrative in the discussion and conclusion sections with the data presented in the graphs now provides a clearer outline of our observations and of implications for double duty research and practice. 

Minor comments:

9. Abstract – the abstract Methods and Findings section should be extended (maybe reducing the background and splitting this in two sections, methods and findings) to give more importance to key messages highlighted by the review.

Response: Done. The abstract was revised to incorporate suggestions from both reviewers. 

REVIEWER 2

1. This study question is of public health interest given the high prevalence of obesity in youth, adolescents and adults around the world and this paper focuses on interventions including a nutrition and physical activity component to reduce obesity.

GENERAL COMMENTS

Abstract:

2. The first paragraph is very confusing.

Response: Done. As above, the abstract was revised to incorporate suggestions from both reviewers.

Introduction:

3. The first couple of paragraphs are quite convoluted and it is difficult for the reader to determine the direction of the authors' message and purpose.

Response: Done. The introduction has been revised. 

4. DDA's are not well explained, not sure of their importance as it relates to the purpose of the review.

Response: Done. We have added further explanations on DDA throughout the text, feasibility of DDA-assessment based on study/programme characteristics (Figure 5), and assessment of DDA eligibility (Supplementary Table 2) based on WHO guidance on double-duty (WHO 2017a, 2017b) and Cochrane Group PROGRESS+ framework (Kavanagh et al. 2008). 

5. The introduction is verbose and quite exhausting to read. It would be beneficial to the reader if the introduction was reduced to 3-4 paragraphs and linearly constructed to lead the reader from the problem to the purpose of the study.

Response: Done. We revised and shortened the introduction, sharpened the focus and harmonised the flow between paragraphs. 

6. In the scientific literature, many reviews have been conducted on the topic of nutrition and physical activity interventions. How is this review different? What does this review contribute to the literature that other studies have not presented?

Response: Whilst many reviews exist on nutrition and physical activity interventions, integrated actions combining nutrition and physical activity and tackling health and wellbeing provide a vantage point for identifying potentially unexploited pathways which have yet to be analysed through a double-duty lens. We believe that this review constitutes an important step in understanding the extent to which this can be accomplished, thus identifying important gaps in the knowledge base for action to tackle multiple forms of malnutrition simultaneously. 

Methods:

7. From what I can tell, the readers did not provide how/if methodological quality was assessed for these studies. How many of theses studies were classified as high quality? moderate quality? low quality? This assessment provides perspective on the level of evidence for the association between nutrition/PA and obesity-related outcomes.

Response: Done. Details of critical assessment were provided in Table form (Supplementary Table 1). The rationale is specified in the main text. Based on evidence that standard grading systems and score-based tools are not appropriate in such multidisciplinary research on emerging issues (CCQMG 2011; DfID 2014; Cornish 2015), due to standard high rating not being attainable for some disciplines and methodologies (DfID 2014), critical appraisal of the studies was intended to ensure better clarity and transparency, rather than quantify quality by attributing summary scores to each study. The Cochrane (CCQMG 2011) rationale which underpins our approach, also reiterated in other development-related evidence appraisal resources, advises against such quality assessment tools, which may be detrimental rather than beneficial when applied to certain types of reviews.

8. Also, while the outcomes are provided, the reader is not informed on the assessment of these outcomes, which are critical to evaluating the level of evidence. Also, more details on the interventions is suggested. Ex: The dose of PA? For 5-year interventions, did the intervention exposure change over time? Were there dissemination aspects to these interventions? It would also be interesting to comments on the participant burden and cost-to-benefit ratio, however that may be beyond the scope of the review or simply added to the discussion would suffice.

Response: Given the variety of evidence and the lack of objectivity of self-reported data used in many of the included studies, this would not be feasible without oversimplification. For this reason, where available we extrapolated hard data, which is presented in greater detail in tabular form (Table 1). However, reporting of quantifiable data is not consistent across studies. The need to define ‘success’ when dealing with diverse study types, and assessment of each study against measures of success, would also be problematic, for the reasons listed in the comment above. 

Results:

9. In most reviews, it is common for authors to provide numeric data. Some is provided here, but what percent were successful? Unsuccessful? The different platforms was an informative layout. I think added more quantitative information would also be helpful.

Response: Same as above. This can and should be done in the case of comparable study designs/methods/reporting but not all studies conventionally included in scoping reviews are suitable. Where data was available, it was reported transparently. However, beyond the reporting of data in tabular form, methods of statistical synthesis are not appropriate for this review. Reasons considered legitimate by the authors - and listed as such by Cochrane (2019) in its advising against statistical synthesis - include limited evidence, incompletely reported outcome/effect estimates, different effect measures used across studies and bias in the evidence, e.g. derived from self-reported data and insufficiently rigorous independent evaluation. These characteristics were highlighted where relevant throughout the manuscript.

---

## [Decision Letter · Decision Letter 1]

15 May 2020

Integrating nutrition and physical activity promotion: a scoping review

PONE-D-19-32917R1

Dear Dr. Casu,

We are pleased to inform you that your manuscript has been judged scientifically suitable for publication and will be formally accepted for publication once it complies with all outstanding technical requirements.

With kind regards,

Mariusz Duplaga, Ph.D., M.D., Ass. Prof.

Academic Editor

PLOS ONE

Additional Editor Comments (optional):

Reviewers' comments:

Reviewer's Responses to Questions

**Comments to the Author**

1. If the authors have adequately addressed your comments raised in a previous round of review and you feel that this manuscript is now acceptable for publication, you may indicate that here to bypass the “Comments to the Author” section, enter your conflict of interest statement in the “Confidential to Editor” section, and submit your "Accept" recommendation.

Reviewer #1: All comments have been addressed

Reviewer #2: (No Response)

2. Is the manuscript technically sound, and do the data support the conclusions?

Reviewer #1: Yes

Reviewer #2: Yes

3. Has the statistical analysis been performed appropriately and rigorously? 

Reviewer #1: N/A

Reviewer #2: N/A

4. Have the authors made all data underlying the findings in their manuscript fully available?

Reviewer #1: Yes

Reviewer #2: Yes

5. Is the manuscript presented in an intelligible fashion and written in standard English?

Reviewer #1: Yes

Reviewer #2: Yes

6. Review Comments to the Author

Reviewer #1: All comments have been extensively addresses.

The manuscrip is now more easy to read and methods are more trasparent.

Reviewer #2: The manuscript has much improved and better highlighted the its importance to the field of physical activity and nutrition.

7. PLOS authors have the option to publish the peer review history of their article (what does this mean?). If published, this will include your full peer review and any attached files.

Reviewer #1: No

Reviewer #2: No

---

## [Editor Report · Acceptance letter]

20 May 2020

PONE-D-19-32917R1 

Integrating nutrition and physical activity promotion: a scoping review 

Dear Dr. Casu:

I am pleased to inform you that your manuscript has been deemed suitable for publication in PLOS ONE. Congratulations! Your manuscript is now with our production department. 

With kind regards,

on behalf of

Dr. Mariusz Duplaga 

Academic Editor

PLOS ONE